# Composition and stage dynamics of mitochondrial complexes in *Plasmodium falciparum*

Felix Evers [1], Alfredo Cabrera-Orefice[2,3], Dei M. Elurbe[3], Mariska Kea-te Lindert[4,5], Sylwia D. Boltryk[6,7], Till S. Voss[6,7], Martijn A. Huynen[3], Ulrich Brandt [2,8] & Taco W. A. Kooij [1✉]

Our current understanding of mitochondrial functioning is largely restricted to traditional model organisms, which only represent a fraction of eukaryotic diversity. The unusual mitochondrion of malaria parasites is a validated drug target but remains poorly understood. Here, we apply complexome profiling to map the inventory of protein complexes across the pathogenic asexual blood stages and the transmissible gametocyte stages of *Plasmodium falciparum*. We identify remarkably divergent composition and clade-specific additions of all respiratory chain complexes. Furthermore, we show that respiratory chain complex components and linked metabolic pathways are up to 40-fold more prevalent in gametocytes, while glycolytic enzymes are substantially reduced. Underlining this functional switch, we find that cristae are exclusively present in gametocytes. Leveraging these divergent properties and stage dynamics for drug development presents an attractive opportunity to discover novel classes of antimalarials and increase our repertoire of gametocytocidal drugs.

[1] Department of Medical Microbiology, Radboudumc Center for Infectious Diseases, Radboud Institute for Molecular Life Sciences, Radboud University Medical Center, Nijmegen, the Netherlands. [2] Radboud Institute for Molecular Life Sciences, Radboud University Medical Center, Nijmegen, the Netherlands. [3] Centre for Molecular and Biomolecular Informatics, Radboud Institute for Molecular Life Sciences, Radboud University Medical Center, Nijmegen, the Netherlands. [4] Electron Microscopy Center, RTC Microscopy, Radboud Institute of Molecular Life Sciences, Radboud University Medical Center, Nijmegen, the Netherlands. [5] Department of Cell Biology, Radboud Institute of Molecular Life Sciences, Radboud University Medical Center, Nijmegen, the Netherlands. [6] Department of Medical Parasitology and Infection Biology, Swiss Tropical and Public Health Institute, Basel, Switzerland. [7] University of Basel, Basel, Switzerland. [8] Cologne Excellence Cluster on Cellular Stress Responses in Aging-Associated Diseases (CECAD), University of Cologne, Cologne, Germany. ✉email: taco.kooij@radboudumc.nl

Malaria parasites harbour only a single, indispensable mitochondrion with a minimalistic mitochondrial DNA (mtDNA) encoding just three proteins: COX1, COX3 and CYTB[1,2]. The latter is the target of the potent antimalarial atovaquone[3]. Its activity on asexual blood-stage (ABS) parasites is not directly mediated by inhibition of the oxidative phosphorylation (OXPHOS) pathway but by blocking ubiquinone regeneration required to sustain de novo pyrimidine biosynthesis[4]. *Plasmodium* gametocyte development and mosquito colonization on the other hand are critically dependent on multiple mitochondrial functions including an active respiration[5–8]. Another remarkable feature observed in the murine malaria model parasite *Plasmodium berghei* is the apparent absence of cristae in ABS parasites[7]. *P. berghei* gametocytes (GCT), however, do possess these inner mitochondrial membrane folds where OXPHOS complexes typically accumulate. Due to high sequence diversity and the poor mitochondrial targeting predictions, much of the *Plasmodium* mitochondrial proteome remains undisclosed. This is painfully illustrated by our limited understanding of a pathway as central to mitochondrial functioning as the OXPHOS pathway. The marked absence of complex I is compensated by a single subunit type II NADH:ubiquinone oxidoreductase. For CII-V, only 23 likely orthologues of 48 canonical components have been identified[9] and no comprehensive biochemical analysis of any of the individual complexes has been done so far. Recent studies in the related apicomplexan parasite *Toxoplasma gondii* have suggested a divergent and unusual composition of cytochrome *c* oxidase (CIV)[10] and $F_1F_O$-ATP synthase (CV)[11]. To this date, only limited data about multiprotein complexes in *Plasmodium* species are available. Function annotations, co-expression patterns, and homology data have been integrated in silico to predict possible protein interactions[12,13]. However, this approach is hampered by the lack of annotated orthologues for many proteins, limited temporal resolution of expression data, and imperfect correlation between transcription and translation timing in *Plasmodium* species[14]. A systematic yeast two-hybrid screen generated protein interaction data for 25% of the proteome[15] but pairwise expression of protein fragments outside of their native context is not necessarily representative or suitable to uncover all relevant protein-protein interactions, especially in the case of multiprotein complexes.

Recent progress in label-free quantitative mass spectrometry (MS) combined with very high sensitivity and speed microscale native fractionation techniques offers the prospect to uncover protein associations by comigration or co-elution[16]. Blue native polyacrylamide gel electrophoresis (BN-PAGE)[17] offers high-resolution separation of intact complexes over a wide mass range without requiring genetic interventions or prior modifications of the sample[16]. This approach, termed complexome profiling, provides the inventory of protein complexes in a single experiment. It has allowed for major advances by finding novel components of OXPHOS complexes and uncovering assembly intermediates and interactions in human[18], plant[19] and yeast[20] mitochondria. A prior study by Hillier et al.[21] has demonstrated that this approach is feasible and effective in *Plasmodium* spp., but due to different scope, sample complexity and comparatively harsh detergent conditions has failed to identify assembled OXPHOS complexes.

Here, we apply complexome profiling to preparations of *Plasmodium falciparum* ABS parasites and GCT enriched for mitochondria. The extensive datasets provide a wealth of information on *Plasmodium* protein complexes allowing the identification of numerous previously suggested and novel components of all OXPHOS complexes. Critically, we uncover stark OXPHOS complex abundance differences between asexual and sexual blood-stage parasites, consistent with the metabolic switch

hypothesis and coinciding with the appearance of cristae as supported by our ultrastructural observations. Further analysis of these parasite-specific OXPHOS components could pave the way towards novel drug targets and enables a better understanding of this divergent and fascinating mitochondrial biology.

## Results

**Stage-specific mitochondrial ultrastructure in *P. falciparum*.** To provide ultrastructural support for the increasing evidence for stage-specific mitochondrial metabolism and function, we performed transmission electron microscopy (TEM) of NF54 wild-type *P. falciparum* ABS parasites and stage V GCT (Fig. 1). Prior TEM-based investigations have demonstrated the presence of cristae in *P. berghei* GCT while ABS parasites were acristate[7]. Cristae in *P. falciparum* stage IV GCT have also been suggested, but low image quality and absence of ABS micrographs, do not allow definitive conclusions[22]. Our data confirm the stage-specific presence of cristae inside the *P. falciparum* mitochondrion (Fig. 1). In ring-stage parasites, the mitochondrion is elongated, acristate and not very electron dense. While significantly larger, the overall appearance is unchanged in trophozoites. In mature schizonts, each daughter merozoite harbours one small, acristate, electron-lucent mitochondrion in close proximity to one four-membrane-bound apicoplast. We also observed that these organelle pairs are distributed to merozoite compartments prior to inclusion of the nuclei (Supplementary Fig. 1). In GCT, clear and abundant internal membranous structures are observed within the mitochondrion, which we assume to be tubular cristae due to their resemblance to tubular cristae observed in steroid-producing[23] cells and *T. gondii*[24]. In addition, the organelle appears more electron dense than in ABS parasites and covers larger distinct areas suggesting an increase in size and level of branching (Supplementary Fig. 2). The multiple mitochondrial sections without apparent connection are assumed to be part of one heavily branched mitochondrion[25] but appear to be distinct as the 3D conformation cannot be appreciated in the 2D micrographs. This obvious discrepancy between all ABS parasites on one side and mature GCT on the other side prompted us to investigate how these changes are reflected at the protein level.

**Complexome profiling of *Plasmodium falciparum*.** Migration patterns of individual proteins were obtained by complexome profiling of mixed ABS parasites and stage V GCT. We employed three different enrichment methods (1: syringe lysis with saponin; 2: syringe lysis without saponin; 3: nitrogen cavitation), two different detergents (D: digitonin; M: n-Dodecyl β-D-maltoside, DDM), as well as two genetic backgrounds were analyzed ("Methods" and Supplementary Table 1 for further details). Samples were named according to the combination of these three parameters and lettering indicating different replicates (Supplementary Table 1). Saponin treatment during shearing was used to test whether specific depletion of relatively cholesterol-rich non-mitochondrial membranes by saponin[26] could improve mitochondrial enrichment. With some notable exceptions, which will be described later, the results were consistent across all enrichment methods. Using the stronger detergent DDM did not lead to detection of more proteins or assembled complexes (Supplementary Table 1) and was consequently omitted in favour of digitonin solubilization for gametocyte samples. To overcome the challenge of obtaining sufficient gametocyte material, we used a recently established inducible gametocyte producer line (NF54/iGP2) (Boltryk et al., in revision[27]) that allows for the synchronous mass production of GCT through conditional overexpression of gametocyte development 1 (GDV1), an activator of sexual commitment[28]. We observed no marked differences in proteome or morphology for ABS and GCT stages between the initial

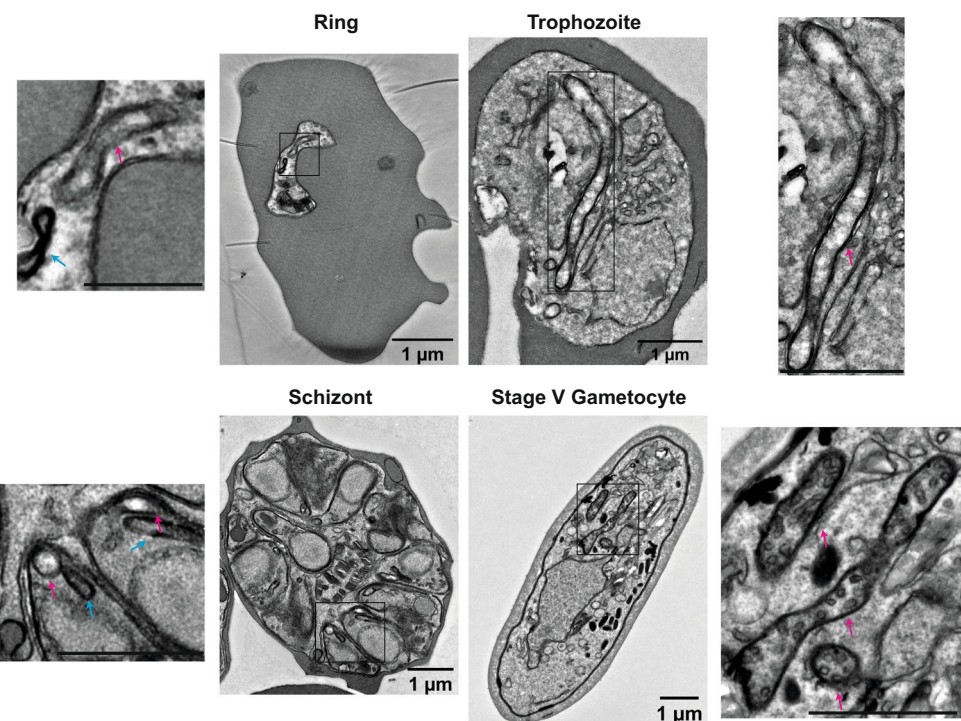

**Fig. 1 Representative electron micrographs of *Plasmodium falciparum* blood stages.** Enlarged sections show ultrastructural differences between the mitochondrion (magenta arrow) in ABS parasites and gametocytes. The mitochondrion presents as an electron-lucent, acristate structure during ABS development, while the gametocyte mitochondrion appears electron-dense and packed with tubular cristae. Also note the close proximity of the four-membrane-bound apicoplast (light-blue arrow) in ring- and schizont-stage parasites. Scale bar ring-stage parasite higher magnification crop, 0.5 µm; all other scale bars, 1 µm. In total, three independent ABS samples and three independent gametocyte samples were imaged and observations were consistent across all.

samples prepared with wild-type NF54 (GCT1Da, ABS1Ma, ABS1Da) and all remaining samples prepared with NF54/iGP2 parasites (Supplementary Table 1). Across all samples (662 fractions) a total of 1759 unique proteins were identified (Supplementary Data 1). All raw and processed data generated in this study were deposited at the ComplexomE profiling DAta Resource (CEDAR; www3.cmbi.umcn.nl/cedar/browse/experiments/CRX23)[29].

It should be noted that abundant proteins from other *P. falciparum* cell compartments were also readily identified. Taking advantage of the latter and to validate the approach, we first investigated whether well-known and previously described complexes could be identified correctly and what the impact was of different isolation methods (Fig. 2).

**Validation with common eukaryotic and *Plasmodium*-specific complexes.** The protein folding/degradation-involved endoplasmic reticulum membrane complex (EMC) genes are conserved across most eukaryotes[30]. We detected comigration of all putative EMC components, with the exception of EMC6, at an apparent molecular mass ($M_r^{app\cdot}$) of ~550 kDa (Fig. 2a). All previous proteomics experiments for which data are available on PlasmoDB (https://plasmodb.org)[31] also failed to detect EMC6 in ABS parasites and gametocytes despite it having similar transcription profiles as other EMC components, indicating its absence or challenging detection by MS. The $M_r^{app\cdot}$ differs from the predicted mass of 294 kDa, possibly due to presence of N-glycans on multiple components[32,33], interaction with other unidentified proteins[34], or a larger oligomeric state. Nevertheless, consistent comigration of the identified subunits strongly indicates the presence of a canonical EMC assembly in *P. falciparum*.

The presence of a proteasome complex represents another universal eukaryotic feature. A prior study has elucidated the

composition and $M_r^{app\cdot}$ of the *P. falciparum* 20S proteasome[35]. We confirmed these results, identifying all 14 subunits of the 20S proteasome comigrating as a clearly defined complex at ~690 kDa and a less abundant, slightly larger assembly (Fig. 2b). Most components of the regulatory 19S particle comigrated in a dominant large and secondary small assembly. The lack of comigration between regulatory components and the core 20S proteasome suggests limited stability of the 26S or 30S assemblies under these conditions. Interestingly, the regulatory subunit 2 (RPN2) seemed to associate with both the dominant 20S and 19S assemblies, while the proteasome activator subunit 28 (PA28) was found exclusively associated with the larger 20S complex. The putative 26S proteasome non-ATPase regulatory subunit 9 (PSMD9) and regulatory subunit 13 (RPN13) were not found comigrating with any of the observed assemblies. We observed that saponin treatment depletes all proteasome-associated assemblies from the respective profiles, suggesting either reduced cytosolic contaminants or a specific detergent-complex interaction upon saponin treatment (Supplementary Fig. 3). Conversely, the studied membrane proteins were not significantly affected under these conditions.

To facilitate waste removal and nutrient uptake through their host cell membrane, malaria parasites have evolved a unique complex composed of the high molecular mass rhoptry proteins (RhopH)[36]. Initially thought to be composed of three subunits, recent studies have implicated additional proteins and estimated its mass as ~670 kDa[37]. Although CLAG3.2 was not detected in any of the samples, presumably due to a high overlap in shared and consequently non-unique peptides with RhopH1, our observations otherwise confirmed the composition and size of this extended RhopH complex and suggested a new component, which we termed RhopH associated protein 1 (RhopHA1; PF3D7_0220200; Fig. 2c). RhopHA1 comigrated consistently

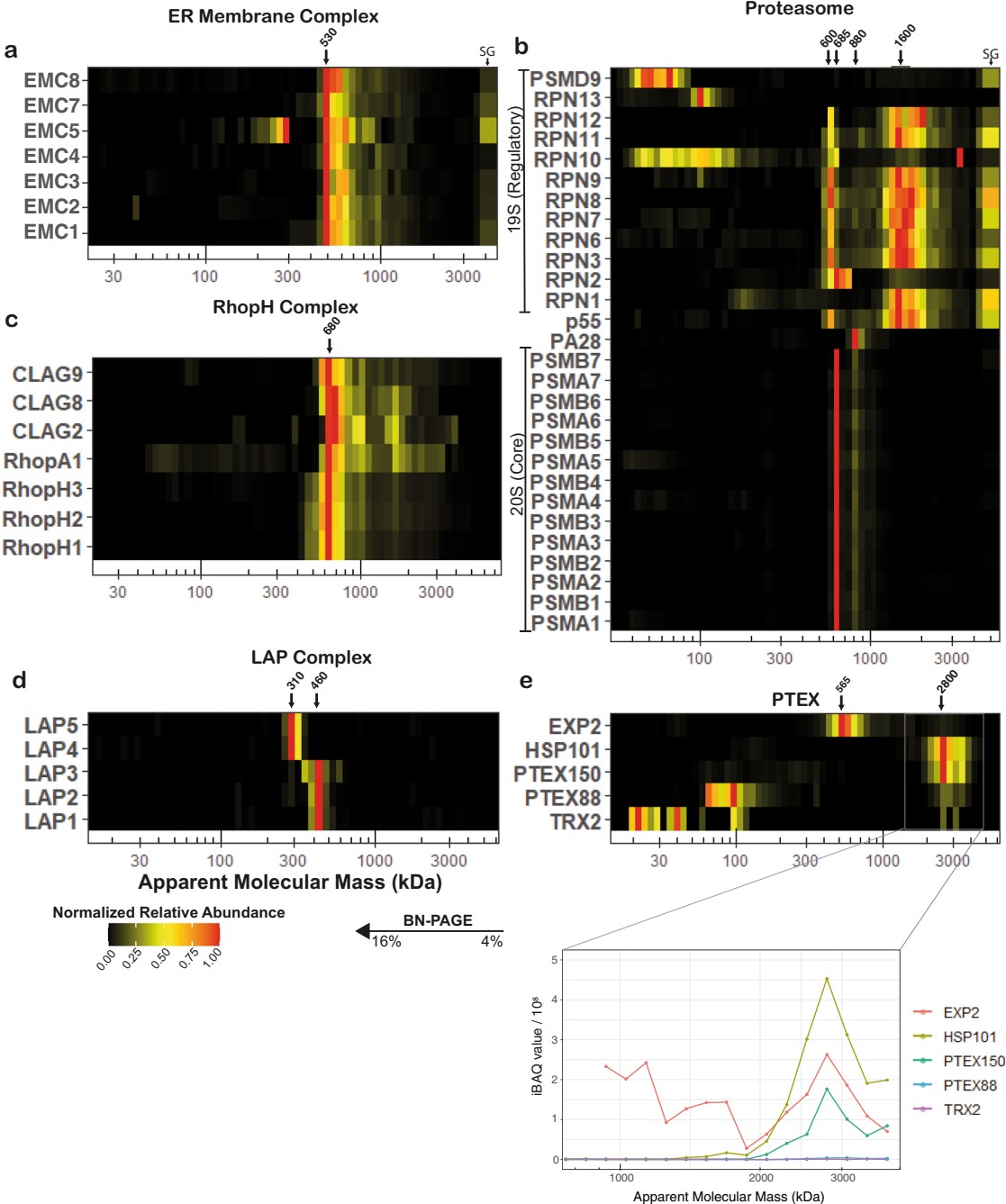

**Fig. 2 Migration profiles of proteins associated with previously described complexes.** Common eukaryotic and parasite-specific protein complexes identified in mitochondrially enriched fractions of *P. falciparum*. **a** Putative EMC components, representative heatmap from sample ABS3D. All detected EMC components comigrate at an $M_r^{app.}$ of ~530 kDa. **b** Proteasome components, representative heatmap from sample ABS3D. 20S components comigrate at an $M_r^{app.}$ of ~690 kDa and to a lesser degree at ~880 kDa. 19S regulatory components comigrate at two distinct sizes, a major proportion at ~1600 kDa and a smaller fraction at ~600 kDa. **c** RhopH complex, representative heatmap from sample ABS1Da. PF3D7_0220200 (RhopHA1) shared RhopH complex pattern consistently and thus was putatively assigned to the RhopH complex. **d** LAP complex, representative heatmap from sample GCT1Da. LAP complex components in gametocytes migrated as two distinct subcomplex consisting of LAP1-3 (~460 kDa) and LAP4-5 (~310 kDa), respectively. In addition, a faint putative assembly intermediate consisting of LAP2 and LAP3 was observed at ~310 kDa. **e** PTEX, representative heatmap (upper panel) and line chart of iBAQ values in the 750–4000 kDa mass range from sample ABS1Da. The PTEX complex including auxiliary subunits can be observed to comigrate at an $M_r^{app.}$ of ~2.8 MDa. Due to the high proportion of EXP2 present as the homooligomeric EXP2 complex at ~565 kDa, membership is only evident when comparing absolute intensity values (lower panel) instead of normalized abundances (heatmap). PTEX88 and TRX2 have much lower intensities in this mass range than core components. SG stacking gel. Corresponding Gene IDs can be found in Supplementary Data 1.

with the previously established components and at the predicted mass under all conditions. It has two predicted transmembrane helices, a PEXEL motif, and no detectable homologues outside the *Plasmodium* genus.

Successful development in the mosquito vector requires expression of six LCCL domain-containing proteins that form a complex in the crystalloid, an organelle unique to *Plasmodium* mosquito stages[38,39]. In *P. berghei*, early assembly of this complex

is prevented through translational repression of LAP4-6 in gametocytes[40]. In *P. falciparum*, LAP1-5 are transcribed from stage II gametocytes onwards and are readily detected by MS, while there is no MS evidence of LAP6 in gametocytes and transcription only commences in stage V gametocytes[41,42]. Immunofluorescence microscopy with one antiserum suggested the presence of LAP6 but at an entirely different localization than the other LAP proteins[39,43]. We did not detect LAP6 in any of the samples but found that in stage V gametocytes LAP1-3 and LAP4-5 formed two distinct subcomplexes (Fig. 2d). We also identified a likely LAP2-3 assembly intermediate. The apparent masses of the complexes resemble the sum of their constituents, suggesting that LAP6 was indeed not present. An interesting explanation could be that instead of repressing translation of LAP4-6 as in *P. berghei*, only LAP6 is repressed in *P. falciparum*, which then functions as the assembly factor bringing the two subcomplexes together after fertilization.

The *Plasmodium* translocon of exported proteins (PTEX) is a malaria parasite-specific protein complex essential for the export of parasite proteins into the host erythrocyte[44,45]. The structure of the core complex consisting of three proteins, EXP2, PTEX150, and HSP101, is now well established[46], while function and even association of the non-essential auxiliary subunits PTEX88 and TRX2 remains less clear[47]. Our data indicated that the majority of EXP2 is present independently from the PTEX (Fig. 2e), consistent with its second proposed function as a homooligo-meric nutrient channel[48]. Reassuringly, when comparing intensity-based absolute quantification (iBAQ) values, the three core components showed comparable intensities (lower panel Fig. 2e). Conversely, the majority of PTEX88 and TRX2 migrated at the predicted size of the monomer, while low-intensity comigration with the core complex was inconsistent across the samples and even replicates, suggesting their presence in only a subset of complexes or a limited association under the experimental conditions used (Supplementary Fig. 4). The fact that complexome profiling helps to distinguish the presence of proteins in different (sub)assemblies, highlights a key advantage over more conventional approaches to investigate interactions of promiscuous components or assess assembly pathways.

**Divergent composition and abundance dynamics of CIII and CIV.** Having validated our complexome profiling approach, we next focussed on the OXPHOS complexes. All canonical components of cytochrome $bc_1$ complex (CIII) with obvious *Plasmodium* orthologues, i.e., CYTB, CYTC1, the Rieske subunit, QCR7, and QCR9, comigrated (Fig. 3a), with the notable exception of *Pf*QCR6 (PF3D7_1426900), that was not detected potentially due to its small size, hydrophobicity, and limited generation of identifiable unique peptides. As observed in plants[49], MPPα and MPPβ also associated with CIII, coupling processing peptidase activity to a structural role in replacing the so-called core proteins. Four additional proteins comigrated consistently with CIII subunits. We identified PF3D7_0306000 as a likely orthologue of QCR8 ($E = 6 \times 10^{-6}$), while the other three proteins, which we termed respiratory chain complex 3 associated proteins 1–3 (C3AP1-3; Table 1), were found almost exclusively in Apicomplexa and lack any detectable sequence homology with characterized proteins (Fig. 4a). In other species, CIII forms a dimer of 470–500 kDa[19,50]. At ~730 kDa, the $M_r^{app.}$ of *P. falciparum* CIII was considerably larger but similar to the 690 kDa expected molecular mass for an obligatory dimer including the newly identified subunits.

So far, only five canonical subunits of *Plasmodium* cytochrome *c* oxidase (CIV) have been identified, i.e., COX1, COX2, COX3, COX5b, and COX6b. COX2 is generally encoded in the mtDNA

but in Apicomplexa and Chlorophyceae the gene has been split in two and relocalized to the nucleus[51]. The resulting protein fragments, COX2a and COX2b, were both retrieved in the complexome profiles. Recent research has shown a highly divergent composition of CIV in *T. gondii*, containing 11 subunits specific to Apicomplexa[10]. Comigration of orthologues of all of these subunits with canonical CIV components confirmed this atypical CIV composition for *P. falciparum* (Fig. 3a). Three proteins that were deemed apicomplexan-specific by Seidi et al.[10] have significant sequence similarity to canonical CIV subunits ($E < 0.01$; Fig. 4b), i.e., PfCOX6A (PF3D7_1465000), *Pf*NDUFA4 (PF3D7_1439600), and *Pf*COX4 (PF3D7_0708700). NDUFA4 was originally identified as a complex I subunit but has later been shown to be a stoichiometric complex IV component[52], which is consistent with our observations. Furthermore, we identified a minimal level ($E = 0.74$) of sequence conservation at the C-terminus of PF3D7_0306500 with COX5C in *Arabidopsis thaliana*, which is orthologous to COX6C from Metazoa and COX9 in fungi. In support of its potential orthology, the conserved residues are all located at the interface of the transmembrane region of COX9 and the other CIV subunits (Supplementary Fig. 6). This part of the protein also appears to contain a glycosyltransferase domain ($E = 1 \times 10^{-28}$), which typically facilitates transfer of a phosphorylated ribose to a substrate protein and is conserved in *T. gondii* and *Vitrella*. Through sequence profile-based searches, the comigrating protein PF3D7_1345300 was identified as orthologous to ApiCOX16 ($E = 4.5 \times 10^{-41}$), which was previously assumed to be *T. gondii*-specific. In addition, we identified five uncharacterized, largely myzozoan-specific proteins that consistently comigrated with the complex and which we termed respiratory chain complex 4 associated proteins 1–5 (C4AP1-5; Table 1).

The complexome profiles suggested staggering abundance differences between ABS parasites and stage V gametocytes (Fig. 3a). Following enrichment by nitrogen cavitation, intensity values for suggested CIII and CIV components are on average 9-fold and 20-fold higher in gametocytes than in ABS parasites. This is not contradicted by the seemingly high relative abundance of ApiCOX18 in the ABS heatmap, since in this case the much lower abundance approached the detection limit causing a normalization artefact. When averaging all digitonin-solubilized samples, stage differences were 6-fold and 23-fold for CIII and CIV components, respectively (Supplementary Table 2).

Finally, to better visualize the presence of higher-order assemblies, we renormalized relative abundances based on intensity values detected at an $M_r^{app.} > 750$ kDa. Thus, we identified complex-specific higher-order assemblies possibly corresponding to, a CIV dimer (Fig. 3c, peak a), the obligatory CIII dimer associating with an CIV monomer (peak b) and dimer (peak d). We also observed putative association of two CIII dimers (peak c). Two additional peaks at 1.9 MDa and 2.3 MDa $M_r^{app}$ (indicated by arrows) could potentially represent larger assemblies of two CIII dimers with one or two CIV monomers, but low relative CIII abundance argues against this assignment. In an attempt to better resolve larger complexes, we repeated the GCT3D experiment but resolved on a 3-16% gel instead and analyzed only the top part of the gel representing $M_r^{app.} > 1000$ kDa (Supplementary Data 2). These HMM (high molecular mass) profiles largely corroborate the previously assigned assemblies (Supplementary Fig. 5a, b) and surprisingly also allowed detection of the previously undetected QCR6 subunit (Supplementary Fig. 5c). It is noteworthy, that the larger putative respiratory supercomplexes peaks are exclusively observed in the gametocyte samples. Even when disregarding absence of distinct peaks, relative intensity at the supercomplex sizes compared to the dominant CIII/CIV bands in ABS parasites, is very low compared

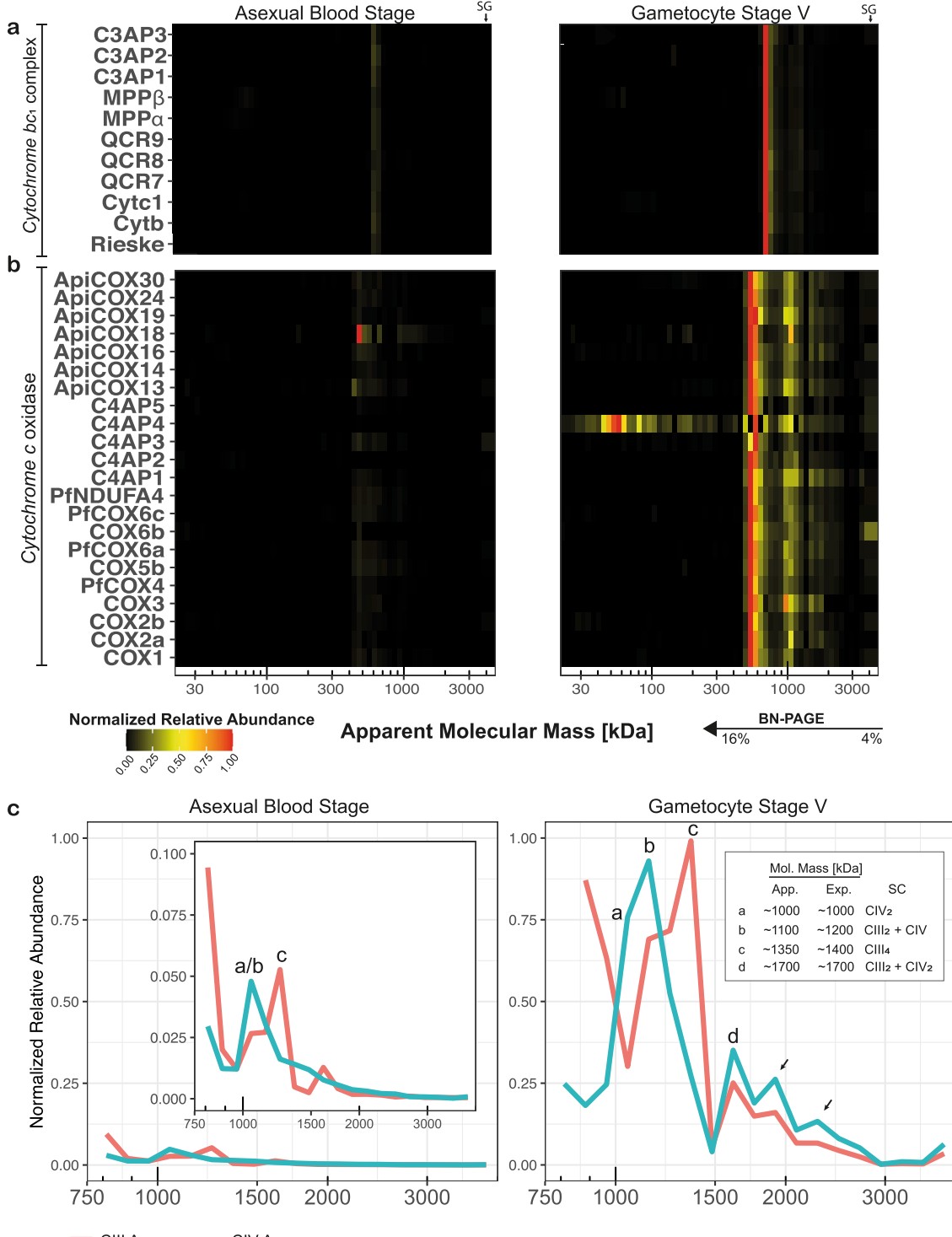

**Fig. 3 Migration and relative abundance of canonical and putatively associated components of respiratory chain complexes III and IV.** An abundance of 1 (red) represents the highest iBAQ value for a given protein between both samples. **a** Heatmap showing comigration of canonical CIII components as well as putative novel components migrating at an $M_r^{app.}$ of ~730 kDa in ABS parasites (left) and gametocytes (right) respectively. **b** Heatmap showing comigration of canonical CIV components as well as putative novel components migrating at an $M_r^{app.}$ of ~570 kDa as well as relative abundance in ABS parasites (left) and gametocytes (right). **c** For detailed analysis of higher-order assemblies, intensity values at $M_r^{app.}$ >750 kDa were renormalized and visualized in a lineplot. Different putative supercomplexes were observed in ABS parasites and gametocytes, denoted with lettering and described in the graph inlet. App. approximate molecular mass based on migration profile, Exp. expected molecular mass based on composition observed in this study, SC supercomplex, $CIII_2$ obligatory CIII dimer, $CIII_4$ association of two CIII dimers, CIV CIV monomer, $CIV_2$ CIV dimer.

**Table 1 Divergent composition of the respiratory chain complexes in *Plasmodium falciparum*.**

| | Name | Gene ID | Mol. Mass[a] | Coverage[b] | MS/MS count[b] | MFS[c] | ToxoGeneID[d] | Tfit[e] | LOPIT[f] |
|---|---|---|---|---|---|---|---|---|---|
| CII | SDHA | PF3D7_1034400 | 70.7 | 29.8 | 219 | −2.83 | TGME49_215590 | −3.96 | mito-soluble |
| | SDHB | PF3D7_1212800 | 378 | 25.2 | 53 | −0.41 | TGME49_215280 | −2.00 | mito-memb |
| | PfSDHC | PF3D7_1448900 | 8.9 | 30.7 | 31 | −2.87 | TGME49_227920 | −2.42 | / |
| | C2AP1 | PF3D7_0808450 | 10.6 | 20 | 10 | / | / | / | / |
| | C2AP2 | PF3D7_1322800 | 14.0 | 25.2 | 30 | 0 | TGME49_252630 | −1.45 | mito-memb |
| | C2AP3 | PF3D7_0109950 | 19.5 | 35.9 | 44 | / | TGME49_315930 | −3.84 | mito-memb |
| | C2AP4 | PF3D7_1346600 | 26.7 | 14 | 21 | −0.26 | TGME49_306650 | −1.80 | mito-memb |
| | Monomer | | | 188 | | | | | |
| | Trimer | | | 564 | | | | | |
| CIII | Cytb | mtDNA | 43.4 | 16.0 | 143 | / | / | / | / |
| | Cytc1 | PF3D7_1462700 | 46.1 | 43.1 | 254 | / | TGME49_246540 | −4.36 | mito-memb |
| | Rieske | PF3D7_1439400 | 41.0 | 59.7 | 646 | −2.58 | TGME49_320220 | −5.76 | mito-memb |
| | QCR6 | PF3D7_1426900 | 11.0 | / | / | −2.49 | TGME49_320140 | −3.69 | mito-memb |
| | QCR7 | PF3D7_1012300 | 23.0 | 78.1 | 355 | −2.62 | TGME49_288750 | −4.04 | mito-memb |
| | QCR8 | PF3D7_0306000 | 17.0 | 38.7 | 95 | −2.85 | TGME49_227910 | −3.20 | mito-memb |
| | QCR9 | PF3D7_0622600 | 11.7 | 53.1 | 118 | −3.1 | TGME49_201880 | −3.68 | mito-memb |
| | MPPalpha | PF3D7_0933600 | 55.7 | 54.5 | 928 | −2.98 | TGME49_202680 | −4.30 | mito-memb |
| | MPPbeta | PF3D7_0523100 | 61.8 | 72.3 | 1345 | −2.75 | TGME49_236210 | −4.74 | mito-memb |
| | C3AP1 | PF3D7_0722700 | 9.5 | 32.9 | 55 | 0 | TGME49_214250 | −3.94 | mito-memb/outlier |
| | C3AP2 | PF3D7_1326000 | 19.3 | 56.4 | 218 | −2.74 | TGME49_207170 | −3.44 | mito-memb |
| | C3AP3 | PF3D7_0817800 | 6.2 | 38.9 | 8 | 0 | TGME49_312940 | −2.27 | / |
| | Dimer | | | 690 | | | | | |
| CIV | COX1 | mtDNA | 57.0 | 7.0 | 41 | / | / | / | / |
| | COX2a | PF3D7_1361700 | 27.3 | 44.3 | 71 | −2.84 | TGME49_226590 | −3.80 | mito-memb |
| | COX2b | PF3D7_1430900 | 19.8 | 52.9 | 96 | −2.53 | TGME49_310470 | −4.18 | / |
| | COX3 | mtDNA | 32.3 | 6.5 | 14 | / | / | / | / |
| | PfCOX4 | PF3D7_0708700 | 27.6 | 26.8 | 200 | −2.42 | TGME49_262640 | −3.49 | mito-memb |
| | COX5b | PF3D7_0927800 | 32.4 | 53.8 | 187 | 0 | TGME49_209260 | −3.07 | mito-memb |
| | PfCOX6a | PF3D7_1465000 | 33.3 | 37.4 | 282 | −2.15 | TGME49_264040 | −2.54 | mito-memb |
| | COX6b | PF3D7_0928000 | 12.2 | 53.4 | 14 | 0 | TGME49_200310 | −0.05 | mito-memb |
| | PfCOX6c | PF3D7_0306500 | 36.4 | 44.1 | 294 | −2.69 | TGME49_229920 | −3.84 | mito-memb |
| | PfNDUFA4 | PF3D7_1439600 | 22.6 | 63.9 | 247 | −2.71 | TGME49_306670 | −3.68 | mito-memb |
| | ApiCOX13 | PF3D7_1022900 | 13.9 | 63.6 | 92 | −2.01 | TGME49_254030 | −4.26 | mito-memb |
| | ApiCOX14 | PF3D7_1339400 | 17.8 | 40.5 | 96 | −2.42 | TGME49_242840 | −3.58 | mito-memb/soluble |
| | ApiCOX16 | PF3D7_1345300 | 10.2 | 63.1 | 65 | −3.07 | TGME49_265370 | 0.65 | mito-memb |
| | ApiCOX18 | PF3D7_0523300 | 16.0 | 56.2 | 42 | −2.92 | TGME49_221510 | −3.28 | mito-memb |
| | ApiCOX19 | PF3D7_1402200 | 22.1 | 39.4 | 78 | −2.67 | TGME49_247770 | −2.61 | mito-memb |
| | ApiCOX24 | PF3D7_1362000 | 23.8 | 39.2 | 114 | −2.19 | TGME49_286530 | −2.82 | mito-memb |
| | ApiCOX30 | PF3D7_0915700 | 22.4 | 60.8 | 192 | −2.96 | TGME49_297810 | −3.64 | mito-memb |
| | C4AP1 | PF3D7_1125600 | 10.0 | 45.8 | 44 | 0 | TGME49_316255 | 0.19 | mito-memb/outlier |
| | C4AP2 | PF3D7_1025800 | 8.2 | 32.8 | 48 | −3.46 | TGME49_263630 | 0.11 | mito-memb |
| | C4AP3 | PF3D7_1003100 | 10.7 | 35.1 | 63 | 0 | TGME49_200310 | −0.05 | mito-memb |
| | C4AP4 | PF3D7_0608400 | 12.7 | 17.9 | 12 | −2.26 | TGME49_312160 | −1.29 | mito-memb |
| | C4AP5 | PF3D7_0809250 | 7.9 | 47.8 | 19 | / | TGME49_225555 | −1.20 | mito-memb |
| | Monomer | | | 476 | | | | | |
| CV | OSCP | PF3D7_1310000 | 30.2 | 15.8 | 4 | −3.12 | TGME49_284540 | −3.94 | mito-memb |
| | ATPα | PF3D7_0217100 | 61.8 | 49.5 | 903 | −2.78 | TGME49_204400 | −3.84 | mito-memb |
| | ATPβ | PF3D7_1235700 | 58.4 | 82.1 | 1741 | −2.51 | TGME49_261950 | −4.84 | mito-memb |
| | ATPγ | PF3D7_1311300 | 35.8 | 12.5 | 12 | −3.03 | TGME49_231910 | −3.94 | mito-memb |
| | ATP*d* | PF3D7_0311800 | 73.5 | 9.9 | 8 | −3.08 | TGME49_268830 | −2.02 | mito-memb |
| | ATP*b* | PF3D7_1125100 | 59.3 | 17.7 | 15 | −3.05 | TGME49_231410 | −5.37 | mito-memb |
| | ATPk | PF3D7_0107400 | 18.8 | 65.4 | 29 | −2.43 | TGME49_260180 | −4.07 | mito-memb |
| | ATP i/j | PF3D7_1360000 | 21.6 | 22 | 6 | −3.11 | TGME49_290030 | −3.88 | mito-memb |
| | ATPδ | PF3D7_1147700 | 17.6 | 71.2 | 19 | −1.7 | TGME49_226000 | −4.57 | mito-memb |
| | ATPε | PF3D7_0715500 | 8.5 | 66.2 | 12 | −1.82 | TGME49_314820 | −3.21 | mito-memb |
| | ATP*a*[g] | PF3D7_0719100 | 21.3 | / | / | −3.6 | TGME49_310360 | −4.49 | / |
| | ATP*c*[g] | PF3D7_0705900 | 18.6 | / | / | −3.39 | TGME49_249720 | −2.98 | / |
| | ATP8 | PF3D7_0934300 | 16.9 | 21.4 | 8 | −2.98 | TGME49_208440 | −3.54 | mito-memb |
| | ATPTG2 | PF3D7_0611300 | 33.5 | 23.2 | 15 | −3.24 | TGME49_282180 | −2.46 | mito-memb |
| | ATPTG3 | PF3D7_0815400 | 18.8 | 28.1 | 8 | −2.76 | TGME49_218940 | −3.92 | mito-memb |
| | ATPTG4 | PF3D7_0905000 | 34.0 | 21.7 | 11 | −3.29 | TGME49_201800 | −4.01 | mito-memb |
| | ATPTG6 | PF3D7_1142800 | 35.5 | 22.2 | 13 | −2.82 | TGME49_223040 | −4.49 | mito-memb |
| | ATPTG9 | PF3D7_1417900 | 16.8 | 18.8 | 16 | −2.4 | TGME49_285510 | −1.87 | mito-memb |
| | ATPTG10 | PF3D7_1024300 | 15.6 | 11.5 | 3 | −3.27 | TGME49_214930 | −1.37 | mito-memb |
| | ATPTG12 | PF3D7_0620100 | 15.5 | 23.8 | 2 | −3.77 | TGME49_245450 | −2.95 | mito-memb |

**Table 1 (continued)**

| Name | Gene ID | Mol. Mass[a] | Coverage[b] | MS/MS count[b] | MFS[c] | ToxoGeneID[d] | Tfit[e] | LOPIT[f] |
|---|---|---|---|---|---|---|---|---|
| ATPTG13 | PF3D7_1303000 | 13.8 | 24.8 | 5 | −2.21 | TGME49_225730 | −3.65 | mito-memb |
| ATPTG15 | PF3D7_0825400 | 14.2 | 28.6 | 7 | −2.39 | TGME49_247410 | −3.90 | mito-memb |
| ATPTG17 | PF3D7_0306600 | 9.9 | 15.3 | 9 | −3.03 | TGME49_310180 | −3.4 | mito-memb |
| Monomer | | 1076 | | | | | | |
| Dimer | | 2152 | | | | | | |

[a]In kDa; estimates based on predicted amino acid composition, no post-translational modifications, cleavage events or lipid association were assumed. For complex mass estimation standard stoichiometry for conserved components (10× ATPc per CV monomer) and 1:1 stoichiometry for novel components was assumed.
[b]Based on all samples.
[c]Mean fitness scores as an indicator of *P. falciparum* gene essentiality[97].
[d]Assigned based on homology searches using HHpred[53] at default settings against *T. gondii* proteome (E < 0.05).
[e]Tfit scores as an indicator of *T. gondii* gene essentiality[98].
[f]Localization estimates of *T. gondii* proteins[99].
[g]ATPa and ATPc were not detected by MS but given their evolutionary conserved nature are expected to form part of the complex.

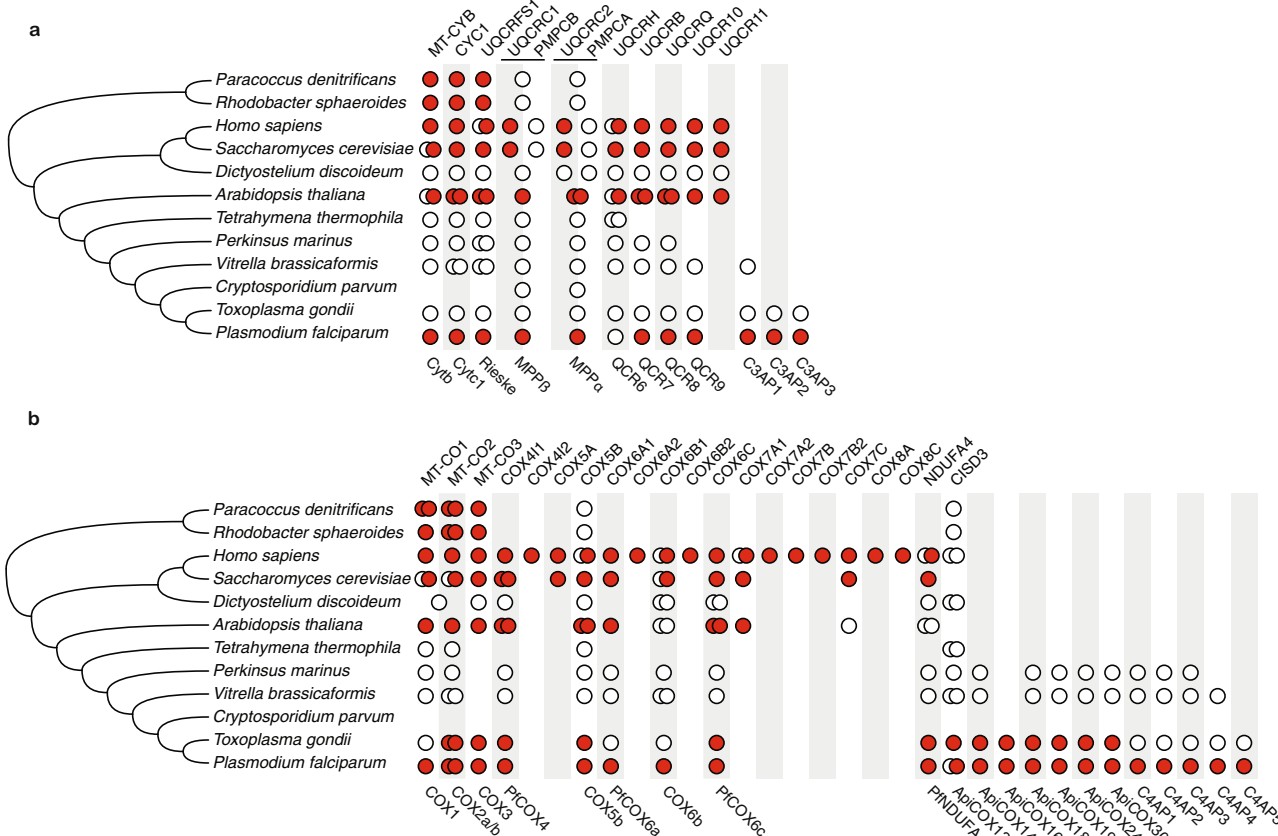

**Fig. 4 Evolution of complex III and IV subunit composition in model species and in the lineage leading to *Plasmodium falciparum*.** The subunit compositions of complex III (**a**) and complex IV (**b**) are based on data from model species and proteins from *P. falciparum* found to comigrate with that complex in this study. Colours depict levels of evidence (red, experimental evidence; white, genomic evidence) linking the subunit to the enzyme. Double circles represent the presence of paralogues and their colour indicates experimental evidence linking them to the complex. For PMPCA/UQCRC2 and PMPCB/UQCRC1, in cases where there has not been a gene duplication, the protein is indicated in the middle between the two columns. Human gene symbols are shown on top; *P. falciparum* gene names are shown below the conservation matrix.

to gametocyes (Supplementary Table 2). This could either be caused by their absence from ABS parasites or falling below the detection threshold due to overall lower abundance of OXPHOS complexes.

**Composition of respiratory chain complexes III and IV in an evolutionary context**. To examine the evolution of *P. falciparum* CIII and CIV in detail, we mapped both the gains and the losses of their respective subunits along an evolutionary tree (Fig. 4). To ensure maximum sensitivity, homology detection was done using HHpred[53] for relatively distantly related taxa for which sequence profiles are available, like mammals and fungi, and using Jackhmmer[54] to map the more recent history of genes among the alveolates.

The three novel CIII proteins (C3AP1-3) have orthologues in *T. gondii*, and one, C3AP1, has an orthologue in *Vitrella brassicaformis*, a sister taxon to the Apicomplexa. They therewith appear to be relatively recent inventions of the Apicomplexa and

their close relatives. Two proteins that are present in CIII from fungi and Metazoa and absent from CIII in *P. falciparum*, i.e., UQCR1 and UQCRC2, have been gained in the evolution of the opisthokonts[55], while UQCRC11 appears to have been lost specifically from the alveolate lineage (Fig. 4a).

Most of the 12 novel proteins we detected in CIV (Fig. 3b) appear to have a myzozoan origin (Fig. 4b). Of the proteins that are absent from *Pf*CIV, COX5A, COX7B, and COX8 appeared relatively recent in the evolution of CIV in opisthokonts[55]. COX7A and COX7C were lost in alveolate evolution. These short proteins have a single transmembrane region that, within the 3D structure of CIV in *Saccharomyces cerevisiae*[56], are in close proximity to each other (Supplementary Fig. 6), suggesting an interlinked alteration on one side of the 3D structure in Apicomplexa. Nevertheless, we cannot exclude that due to sequence divergence they cannot be detected using sequence-based homology detection.

Except for the presence of orthologues in other species, we also examined addition/loss of protein domains in conserved complex members. With respect to CIII, *Pf*CYTC1 contains, relative to CYTC1 in human, an N-terminal extension of ~150 amino acids that is notable because it is specific to Apicomplexa and is present as an individual protein in *Cryptosporidium muris* (CMU_009920) that lacks a traditional mitochondrion[57]. PF3D7_0306500 encodes a 299 amino acid protein of which only the C-terminal ~50 residues are homologous to COX6C. If, as suggested by homology, PF3D7_0306500 interacts with the other members of CIV in the same manner as COX6C, most of the protein would be located in the matrix. Finally, we detected one novel CIV subunit (ApiCOX13) containing a CCCH zinc finger domain likely harbouring a [2Fe-2S] cluster. ApiCOX13 is orthologous to the human mitochondrial matrix protein CISD3, which has been suggested to play an important role in homeostasis of iron and reactive oxygen species[58]. Despite this direct evolutionary relationship, there are some important differences. CISD3 contains two CCCH zing finger motifs of which only one is conserved in ApiCOX13. Furthermore, ApiCOX13 contains a (predicted) C-terminal transmembrane helix, with a (predicted) topology that puts most of the protein in the mitochondrial matrix, while CISD3 is not a transmembrane protein. When in evolution CISD3 has become part of CIV is not known. The C-terminal transmembrane helix can be detected within Apicomplexa and in *V. brassicaformis*, but not in ciliates, suggesting it originated, like many new CIV proteins in *P. falciparum*, in Myzozoa.

**F₁Fₒ-ATP synthase—complex V**. The classical mitochondrial function includes harnessing energy in the chemical bonds of ATP, a process predominantly executed by CV. In the samples processed with methods 1 and 2 (Supplementary Table 1), we were unable to detect any of the predicted $F_1F_o$-ATP synthase components, including putative apicomplexan-specific subunits recently identified in *T. gondii*[11,59]. The notable exceptions were free forms of ATPα and ATPβ without any apparent interaction partners. We suspected that this could have been due to harsh lysis conditions, insufficient quantities of mitochondrial protein, inability of the assembled complex to enter the gel or depletion of the complex through saponin treatment[60]. To address these issues, larger amounts of ABS parasites or gametocytes were lysed through nitrogen cavitation without saponin (method 3; Supplementary Table 1). To detect protein assemblies >5 MDa, the stacking gel was also analyzed. Thus, we found 14 proteins that are associated with CV, either through homology to previously identified *T. gondii* components or through homology to canonical CV components. These CV subunits comigrated at a size of ~2.2 MDa or remained stuck at the interface of the stacking gel and the sample slot in both ABS parasites (Supplementary Fig. 7)

and gametocytes (Fig. 5a). The fraction unable to enter the gel potentially represents higher oligomeric CV states, while we interpret the ~2.2 MDa band as the unusually large CV dimer, both of which were recently identified in *T. gondii*[61]. This is based on a stoichiometry of 10 ATPc subunits for the c-ring, canonical stoichiometry for orthologues to known components and single copies of novel components, predicting a monomer mass of ~1 MDa (Table 1). This, along with the recent elucidation of the ATP synthase structure in *T. gondii*[61] and similar observations in a recent *T. gondii* complexome profile[62], suggests that the ~1 MDa complex observed in other studies probably represent the monomer rather than the dimer or is based on flawed mass calibration[7,59,63] (Supplementary Fig. 9). The gametocyte HMM profiles we deployed to investigate the CIII and CIV super-complexes, permit separation of protein complexes of up to 10 MDa[64], which should allow detection of potential higher oligomeric states of CV. Indeed, we observed comigration of CV subunits beyond the dominant ~2.2 MDa dimer peak (Fig. 5b), however, this migration pattern was not shared across all subunits or resulting in clearly defined peaks at the expected tetramer (4.4 MDa) or hexamer (6.6 MDa) sizes. Despite the limitations of using iBAQ values to determine exact differences in protein abundance, the marked difference between average subunit abundance in CV and CIII/CIV supercomplexes (Supplementary Fig 5b) suggests that CV is present at a relatively low level in sharp contrast with bovine heart mitochondria where CIII and CV are present in similar levels[65]. Nevertheless, our findings suggest the presence of larger CV assemblies and the HMM profiles also allowed an improved detection, increasing identification to 20 putative CV subunits instead of the 13 subunits that were identified in the full profiles.

**Succinate dehydrogenase—respiratory chain complex II**. Succinate dehydrogenase couples succinate oxidation as part of the citric acid cycle to the reduction of ubiquinone in the OXPHOS pathway. CII is generally composed of at least four different subunits: the hydrophilic SDHA and SDHB subunits catalysing succinate oxidation and the hydrophobic SDHC and SDHD subunits anchoring the complex in the inner mitochondrial membrane and providing the binding pocket for haem and ubiquinone. Similar to CV, we were unable to find an assembled CII using methods 1 or 2. In *Plasmodium*, only SDHA and SDHB are experimentally verified[66]. Using method 3, we found comigration of SDHA and SDHB at an $M_r^{app}$ of ~530 kDa (Fig. 5a). Two previously suggested candidates for SDHC (PF3D7_0611100) and SDHD (PF3D7_1010300)[67] were not comigrating with this complex in any of the samples (Fig. 5a). Instead, we identified five putative subunits sharing a common dominant band, although the individual migration patterns were quite heterogeneous and spread over multiple slices. We assigned one candidate as a putative *Pf*SDHC (PF3D7_1448900) as it contains a "DY" motif at positions 52-53 that is conserved in SDHC in a large number of species[68] and of which the tyrosine binds ubiquinone in yeast[69] (Supplementary Fig. 8). However, it contains no recognizable haem-binding motif and only a single predicted transmembrane helix, in contrast to three in *S. cerevisiae* SDHC. The other components we named respiratory chain complex 2 associated proteins 1–4 (C2AP1-4; Table 1), one of which (PF3D7_0808450) is myzozoan-specific and has been shown to play a critical role in ookinete mitochondria in *P. berghei*[70]. Under native conditions, CII can be found as a trimer in prokaryotes[71,72]. The subunit composition suggested in this study predicts a molecular mass of 188 kDa per CII monomer and 564 kDa for the trimer (Table 1) approximating the observed $M_r^{app}$ of 530 kDa. As detection of CII components was limited to ABS3D and GCT3D

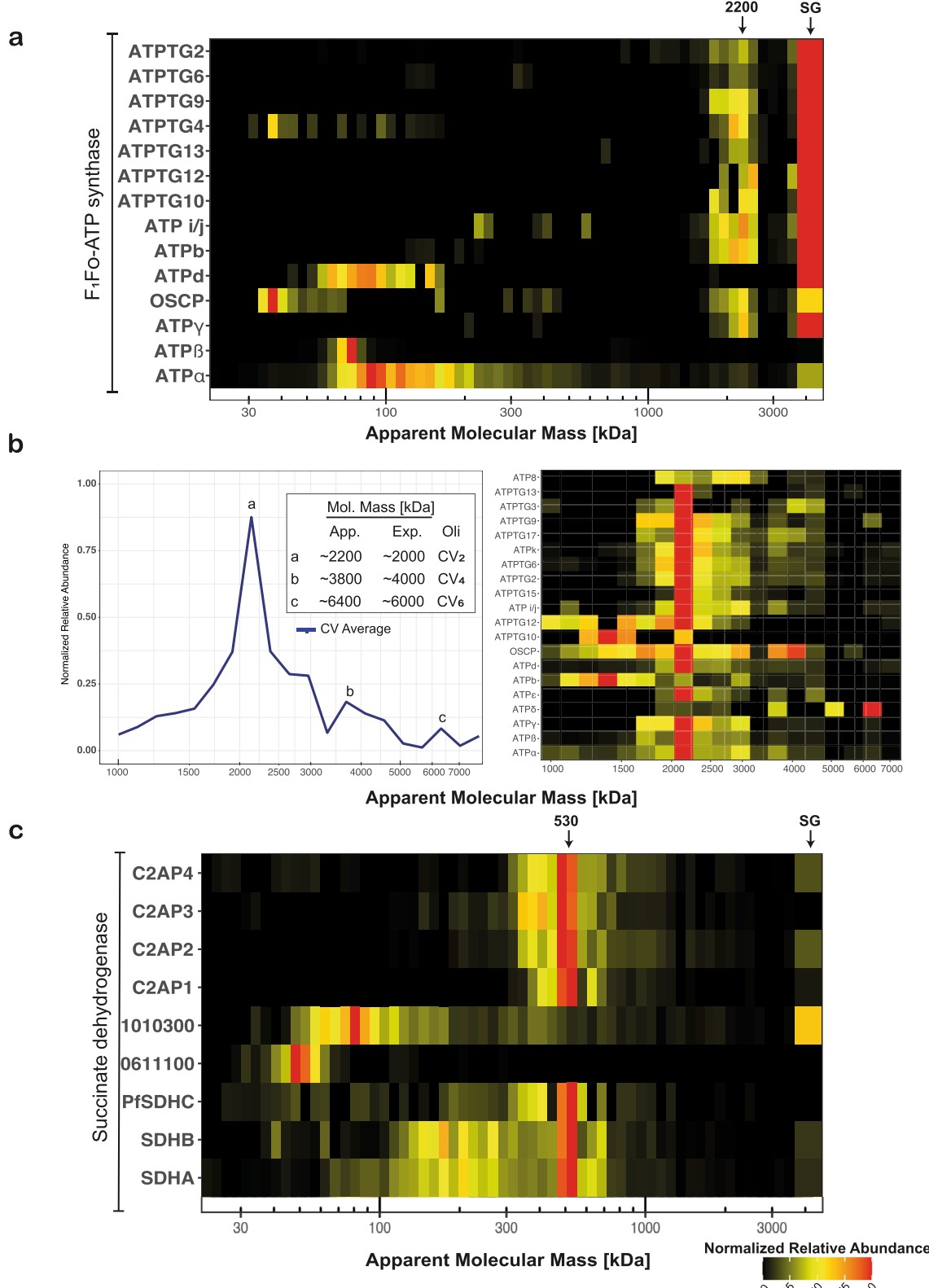

**Fig. 5 Composition and apparent molecular mass of succinate dehydrogenase (CII) and ATP synthase (CV) in *Plasmodium falciparum* gametocytes.**
**a** Heatmap showing migration patterns of canonical ATP synthase components as well as components identified in *T. gondii* in sample GCT3D[59]. Most components show comigration at a size band from 2–3 MDa as well as abundance in the stacking gel interface (rightmost slice). ATPd, ATPβ, OSCP and ATPα appear to be most abundant at their respective monomeric sizes. **b** Heatmap and lineplot showing migration pattern of putative ATP synthase components in 3-16% MDa profiles. Inlet of lineplot describes putatively assigned di- (a), tetra- (b), and hexamer (c) states. **c** Heatmap showing SDHA and SDHB comigrating at an M$_r^{app.}$ of ~530 kDa along with a group of putative novel components. Previously assigned proteins (PF3D7_1010300 and PF3D7_0611100[67]) were not found to be comigrating. Stacking gel (SG) is represented broader in the heatmap and indicated with a black arrow.

(Supplementary Table 1), sample size is small compared to other complexes discussed in this study. In addition, as complexome profiling provides no direct evidence of interaction and only one comparable study in *T. gondii*[62] has suggested a similar composition, further studies will be needed to verify our findings.

**Protein dynamics are in line with a significant metabolic shift in *Plasmodium falciparum* gametocytes.** Metabolomics approaches have indicated a shift in carbon metabolism in gametocytes from anaerobic glycolysis towards increased TCA cycle utilization and presumably increased respiration[73,74], which is also reflected in a general increase of mitochondrial proteins and specifically of TCA proteins in gametocytes[75]. Likewise, this is supported by an increased sensitivity of gametocytes to TCA cycle inhibition[5] and reliance on wild-type cytochrome *b* for transmission[6], which both are non-essential during ABS. A further indication is the de novo appearance of cristae in gametocytes, as they typically serve as hubs for respiration (Fig. 1)[76]. We examined whether this mitochondrial phenotype would also be reflected in the abundance of OXPHOS complexes.

When normalizing the complexome profiles for total intensity of protein detected in a given fraction, a general trend was observed that proteins associated with CIII and CIV were more abundant in gametocytes than ABS parasites by a large margin (Fig. 3, Supplementary Table 2). However, among different gametocyte samples, abundances of OXPHOS associated proteins also varied up to two-fold even after correction. This indicated that the degree of mitochondrial enrichment was not entirely consistent across the different samples, necessitating a more unbiased approach using stage V gametocyte and mixed ABS parasite whole-cell lysate under denaturing conditions. We performed MS analysis on 30 slices of SDS-gel separated lysate, minimizing false-positive identifications by only evaluating the slices that match the predicted molecular mass of the protein (Supplementary Data 3). The obtained results complemented and supported essentially all observations made using complexome profiling (Fig. 6). Compared to ABS parasites, the average abundance levels of OXPHOS complex components in gametocytes were higher 15-fold for CII, 36-fold for CIII, 44-fold for CIV, and 32-fold for CV. This phenomenon included all but one of the putative novel components, further supporting their association with the respective complexes. Outliers were PFSDHN3 from CII, CYTB from CIII, and COX2a from CIV, which showed a comparatively higher abundance in ABS parasites. For subunit COX2a, one identified peptide with a high error and inconsistent migration pattern was observed. After manual removal of this peptide and recalculation, the iBAQ value was in line with the other CIV components. For the other two proteins no obvious outliers were observed at the peptide level. Therefore, these data do not support CII membership of C2AP3. It is noteworthy that all outliers were proteins detected with a low peptide count and few MS/MS events, suggesting decreased reliability when attempting to quantify proteins close to the detection limit.

To further analyze whether this trend was indicative of a larger metabolic shift, we also investigated abundance dynamics of other proteins involved in central energy metabolism (Fig. 7). Complexome (Supplementary Fig. 10) and SDS profiles (Fig. 7a) indicated that enzymes involved in the glycolysis pathway are much more prevalent in ABS parasites. Interestingly, an alternative lactate dehydrogenase (altLDH; PF3D7_1325200)—or potentially malate dehydrogenase as substrate specificity cannot be deduced from sequence—appears to be gametocyte-specific. This is also true for an alternative phosphofructokinase (altPFK; PF3D7_1128300) that appears to have lost crucial residues required for its

function[77]. Stage differences varied for enzymes of the TCA cycle, potentially suggesting different inputs, bottlenecks or alternative utilization of individual enzymes between stages (Fig. 7b), however, these did not correlate with gene essentiality[5]. Alternative ubiquinol-producing enzymes that feed into OXPHOS were also increased in gametocytes but to a lesser degree for DHODH, which had a comparatively higher abundance in ABS parasites (Fig. 7d). DHODH is expected to be abundant in ABS parasites due to their reliance on de novo pyrimidine biosynthesis[4]. All mitochondrial enzymes were found to be comparatively more prevalent in ABS parasites when assessed through mitochondria-enriched complexome profile data (Supplementary Fig. 10). This is possibly due to higher relative mitochondrial content in those samples as is suggested by the comparatively higher abundance of mitochondrial "household" genes VDAC, TOM40, TIM50 and HSP60 (Fig. 7c). Taken together these data support previous metabolomics-based suggestions of a switch towards respiration and away from anaerobic glycolysis in *P. falciparum* gametocytes[73].

## Discussion

We demonstrated the utility of complexome profiling to address the considerable knowledge gap regarding multiprotein assemblies and supercomplex formation in *P. falciparum*. We identified putative novel OXPHOS complex components, suggested a mechanism for regulation of crystalloid formation in *P. falciparum*, and validated and predicted additional features for previously characterized complexes. In addition, we utilized the label-free quantification data from complexome profiling and denatured whole-cell lysates to assess abundance changes between ABS parasites and mature gametocytes. To place our data into an evolutionary context, we integrated phylogenetic analysis allowing us to devise novel hypotheses and assess significance of observed phenomena.

For the novel uncharacterized and apicomplexan-specific OXPHOS complex subunits, it is challenging to estimate biological significance or function. With the exception of ApiCOX13, of which the human orthologue binds iron-sulfur clusters, and the addition of a glycosyl transfer domain to PfCOX6C the new CIII and CIV subunits do not have detectable homologues outside the Myzozoa or indeed any recognizable functional domains that could give us a hint about their potential function. A large proportion of them have predicted transmembrane helices, specifically all three new CIII subunits and nine of the twelve new CIV subunits (Supplementary Table 3). We can speculate they surround CIII and CIV in the membrane plane as has been observed before for supernumerary OXPHOS subunits in opisthokonts[56]. Two of the new subunits might even be functional and structural replacements of the two missing subunits of CIV. However, as we have observed in the evolution of CI in Metazoa, evolutionary new subunits do not necessarily occupy the same location in the complex where subunits are missing[78].

Nevertheless, the observed complexes allow us to draw some important conclusions and raise interesting questions. CIII provides a clear example where *Plasmodium* species resemble plants by both using MPPα and β as structural components[49] unlike animals and fungi where MPPα and β have been replaced by the homologous subunits core 1 and 2 that do not have general MPP activity. The fact that the mitochondrial processing peptidases are tied to the structurally essential core 1 and 2 subunits of cytochrome $bc_1$ complex, presents an interesting trade-off in the context of *Plasmodium* biology. *P. falciparum* ABS parasites are not reliant on OXPHOS outside of ubiquinone recycling for pyrimidine biosynthesis[4], while gametocytes heavily rely on it for successful colonization of and development in the mosquito

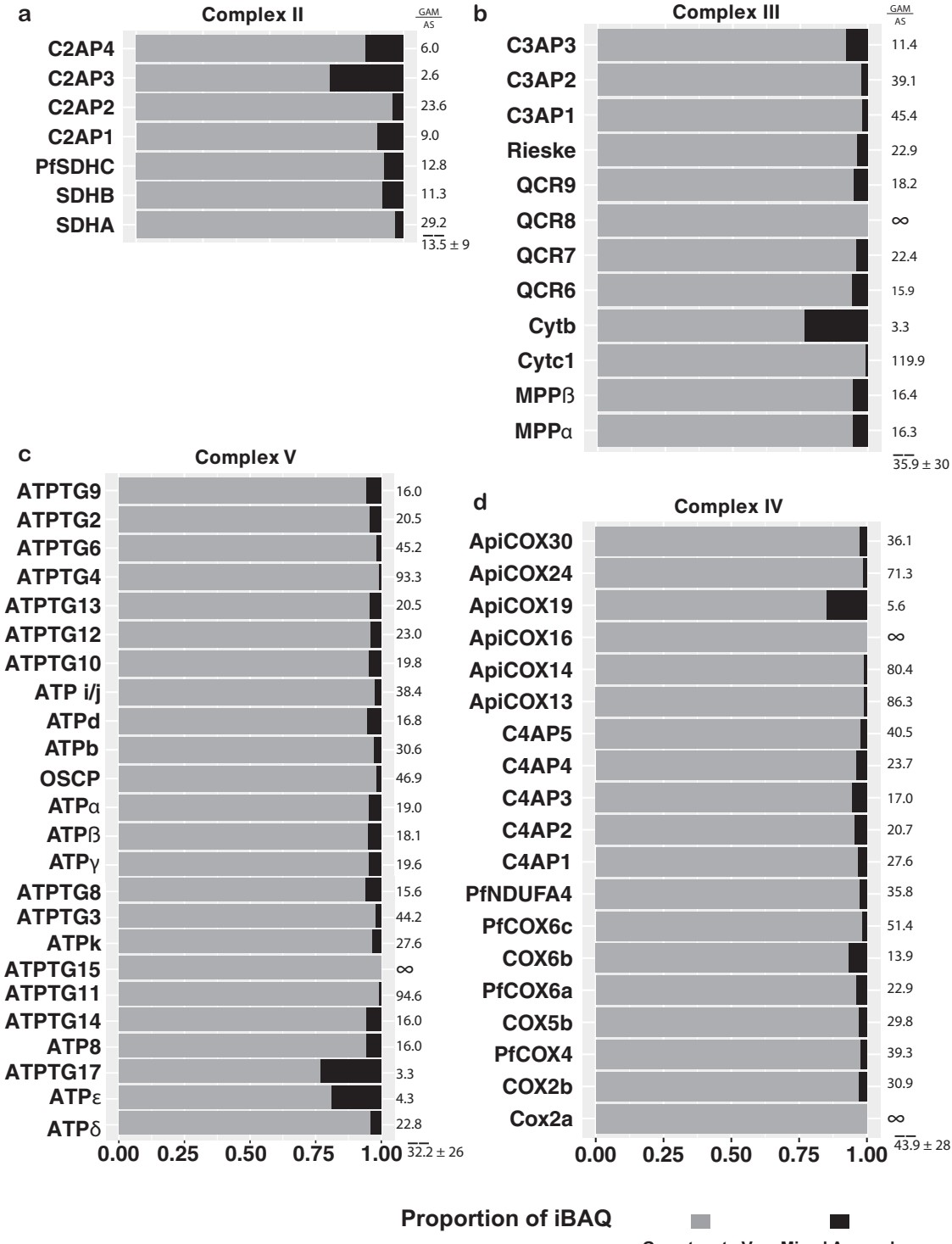

**Fig. 6 Relative quantification of respiratory chain complex components.** Relative abundance expressed in proportion of iBAQ of OXPHOS components in ABS parasites (black) and gametocytes (grey). Data are based on denatured whole-cell lysates separated by SDS-PAGE and analyzed by label-free quantitative MS. Fold differences are indicated next to each bar and for each complex a mean ± SD are indicated below dashed lines. Infinite fold changes were arbitrarily treated as 100 for mean/SD calculations. **a** Putative components of CII. **b** Putative components of CIII. **c** Putative components of CV. **d** Putative components of CIV.

host[6,7]. As a direct consequence, we see a much lower specific content of OXPHOS complexes in ABS parasites compared to gametocytes (Fig. 6). However, presumably ABS parasites still have a comparably high need for MPP activity to facilitate mitochondrial function outside of respiration, as exemplified by putative MPPβ inhibitors showing promise as antimalarials[79,80].

This would necessitate synthesis of the whole complex, which is much less efficient and more challenging from a regulatory standpoint than generating the heterodimer observed in the mitochondria of its host. A dual localization of the processing peptidases as a soluble heterodimer would circumvent this but is not observed under our conditions.

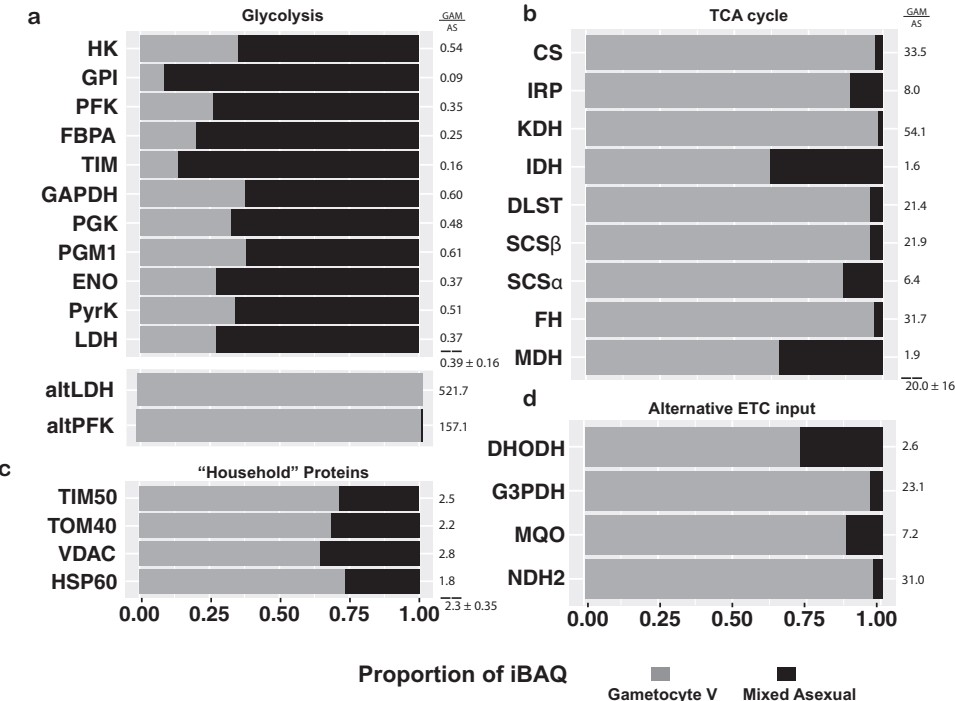

**Fig. 7 Abundance comparison of energy metabolism-related proteins.** Relative quantification of a selection of energy metabolism enzymes (**a**, **b**, **d**) and mitochondrial household proteins (**c**) in ABS parasites (black) and gametocytes (grey) based on denatured whole-cell lysates separated by SDS-PAGE. Fold changes are indicated next to each bar and averages with standard deviation for a group are indicated below dashed lines. Corresponding Gene IDs can be found in Source data.

Similarly puzzling is the apparent increase in size and number of components of OXPHOS complexes, despite the minimalistic mitochondrial genome and single subunit type II NADH:ubiquinone oxidoreductase replacing CI. While it has been shown that migration in BN-PAGE is also influenced by factors other than molecular mass such as ability to bind Coomassie dye, lipid association and intrinsic negative charges[64], our mass estimations do match expected mass of subunit composition observed in this study. In addition, mass estimations are based on bovine heart OXPHOS complexes, which should be representative and have comparable electrophoretic properties to OXPHOS complexes in *P. falciparum*. Conversely, this suggests that for complexome profiling of other compartments, a different set of proteins might be more representative of the electrophoretic properties of proteins of interest.

In most commonly studied eukaryotes, a typical $M_r^{app}$ is ~500 kDa for a CIII dimer, ~210 kDa for a CIV monomer, ~120 kDa for a CII monomer, and ~600 kDa for a CV monomer. Our data suggest respective size increases of around 50%, 130%, 50%, and 70% in *P. falciparum* as compared to complexes III, IV, II and V of more commonly studied eukaryotes. While enlarged OXPHOS complexes have been observed before[55], the size increases in *P. falciparum* are remarkably large. These increases occurred roughly at the time of the origin of multicellular clades, i.e., before the origin of the Myzozoa ~1300 million years ago, or considerably later in evolution before the origin of the Apicomplexa ~900 million years ago[81]. In contrast, fungi and Metazoa have comparably few of such taxon-specific supernumerary subunits. Without further experimental investigation, it is challenging to say whether these size increases correspond to additional functions of the OXPHOS complexes such as the earlier discussed MPP activity. An alternative explanation could be that the transfer of genes originally found on mtDNA to the nucleus necessitated amino acid changes to facilitate import into the

mitochondrion, which in turn required incorporation of additional subunits to maintain the same functionality. Although this is a tempting hypothesis, no strong evidence for it has been obtained from the analysis of the evolution of the mitochondrial genome in conjunction with the mitochondrial proteome in other evolutionary lineages[78,82]. Finally, there also could be a lack of energetic constraints that are afforded by the parasitic lifestyle, which may have allowed the passing on of such bulky complexes. A comparable phenomenon was shown in lower mitochondrial quality control and selective constraints of flightless birds compared to their flying counterparts that retained this energetically demanding ability[83]. This theory, however, is not entirely compatible with the observation that colonization of the mosquito host requires efficient respiration and represents a major bottleneck in the *Plasmodium* life cycle[6].

At this point, it is challenging to suggest a role the putative respiratory supercomplexes might play in *Plasmodium* biology. In mammalian systems, it has been suggested that supercomplexes play an important role in reduction of mitochondrial reactive oxygen species formation and to facilitate more efficient substrate channelling. They have also been suggested to play a role in preventing random protein aggregation in the protein rich inner mitochondrial membrane. However, all these suggested roles have been challenged and no clear consensus has emerged[84]. While our data suggest that the free form respiratory complexes are the dominant species in *P. falciparum* mitochondria, it is hard to estimate whether this balance is representative of the in vivo abundances and what impact solubilization and electrophoresis procedures have on stability of the supercomplexes. Though the low detection levels in ABS parasites do not allow any firm conclusions, the apparent abundance of supercomplexes in gametocytes could hint at their importance for a more efficient functioning. One conclusion that can be drawn though, is that cristae are not a prerequisite of supercomplex formation, though

conversely OXPHOS supercomplexes may be important for forming or maintaining cristae.

In the context of de novo cristae biogenesis in gametocytes, possible consequences of changes in the abundance of CV dimers and their markedly different composition are particularly notable: rows of CV dimers[85] are known to shape cristae by bending their membrane[86]. The observed decrease to ~3% of CV components in ABS compared to gametocytes (Fig. 6) thus provides a straightforward explanation for the absence of cristae in the former. During revision of this manuscript an impressive study was published showing that in *T. gondii*, CV associates into hexamers, which determine cristae shape[61]. This hexamerization is facilitated by at least one non-canonical subunit that is also encoded in *the P. falciparum* genome, suggesting that the large abundance of CV components stuck at the stacking and detection of subunits at a $M_r^{app} > 2.2$ MDa seen in this study, represents the same phenomenon (Fig. 5a, b).

In summary, our comprehensive comparative analysis of the ABS and gametocyte complexome profiles has revealed abundant clade-specific novelties and overwhelming stage differences. A further fundamental understanding of these differences could help to leverage the heavy reliance of gametocytes on unusual and highly divergent mitochondrial complexes as much sought-after gametocytocidal drug targets. To this end, application of genetic tools will be a crucial next step to assess the role and significance of the divergent features proposed in this study. Above all, complexome profiling of *P. falciparum* mitochondria revealed peculiar new biology and fascinating insights in the evolution of eukaryotic respiration and how the malaria parasite has adapted to different environmental challenges at the level of multiprotein complexes.

## Methods

**Parasite culture**. *P. falciparum* strain NF54 was maintained in RPMI [7.4] supplemented with 10% human serum and 5% haematocrit using standard culturing technique[87]. NF54/iGP2 strain was additionally supplemented with 2.5 mM D-(+)-glucosamine hydrochloride (Sigma #1514) for maintenance. For induction, parasites were synchronized with 5% sorbitol[88] and glucosamine was omitted from the medium for 48 h. From day 4–8 gametocyte cultures were treated with 50 mM N-acetylglucosamine (Sigma #A6525) to eliminate ABS parasites[89]. For WT-NF54 gametocyte induction the same procedure was followed except that instead of glucosamine induction, parasites were overgrown for 5 days with only medium exchanges every 48 h prior to N-acetylglucosamine treatment. At day 14 after induction, gametocytes were magnet-purified according to the procedure described previously[90]. ABS parasites were not further enriched prior to processing. Infected red blood cells were freed from host material through 10 min incubation in 10× pellet volume of 0.05% (w/v) saponin in phosphate-buffered saline (PBS; pH 7.4) and subsequent centrifugation at $3000 \times g$ for 5 min. The parasite pellet was washed twice with PBS and the dry pellet was flash frozen and stored at −80 °C.

**Transmission electron microscopy**. For electron microscopy analysis of mitochondria across different ABS parasites and stage V gametocytes, infected red blood cells were fixed in 2% glutaraldehyde in 0.1 M cacodylate (pH 7.4) buffer overnight at 4 °C, washed and cell-pellet was resuspended in 3% ultra-low-gelling agarose, solidified and cut into small blocks. Agarose blocks with cells were post-fixed for 1 h at RT in 2% osmium tetroxide and 1.5% potassium ferrocyanide in 0.1 M cacodylate buffer with 2 mM CaCl2, washed in MQ and incubated in 0.5% thiocarbohydrazide solution for 30 min at RT. After washing agarose blocks with cells were again fixed in 2% osmium for 30 min at RT, washed and placed in 2% aqueous uranyl acetate overnight at 4 °C. After washing agarose blocks with cells were placed in lead aspartate solution (pH 5.5) for 30 min at 60 °C, washed, dehydrated in an ascending series of aqueous ethanol solutions and subsequently transferred via a mixture of aceton and Durcupan to pure Durcupan (Sigma) as embedding medium. Ultrathin sections (80 nm) were cut, air dried and examined in a JEOL JEM1400 electron microscope (JEOL) operating at 80 kV.

**Mitochondrial enrichment**. On the day of the experiment, parasite pellets were resuspended in ice-cold MESH buffer (250 mM sucrose, 10 mM HEPES, 1 mM EDTA, 1× cOmplete™ EDTA-free Protease Inhibitor Cocktail (Sigma), pH 7.4) and washed by centrifugation at $3500 \times g$; 10 min, 4 °C. The mitochondria-enriched fractions were obtained following three different methods. For *methods 1 and 2*, the parasite pellets were resuspended in ice-cold MESH buffer supplemented *with or*

*without* 0.5% (w/v) saponin, respectively, and lysed by 20 strokes through a 27 G needle. Rough debris and unbroken cells were pelleted by low-speed centrifugation ($600 \times g$, 10 min, 4 °C). The supernatant was transferred into a new tube and low-speed centrifugation was repeated. The supernatant was recovered again and centrifuged at a higher speed ($22,000 \times g$, 15 min, 4 °C). The supernatant (cytosolic fraction) was discarded and the pellet (mitochondria-enriched fraction) was resuspended in MESH buffer and kept on ice until usage. For *method 3*, nitrogen cavitation was used for cell disruption. The parasite pellets were washed once in ice-cold MESH, pooled in a total volume of 1 ml and then added to a pre-chilled cell disruption vessel (#4639 Parr Instrument Company). The vessel was pressurized and equilibrated with nitrogen gas at 1500 psi for 10 min on ice. The parasite cells were then sheared through a slow release by nitrogen cavitation. The mitochondria-enriched fraction was obtained by differential centrifugation as described above. Protein concentration was determined using the Pierce™ BCA Protein Assay Kit (Thermo Scientific) using bovine serum albumin as standard.

**Blue native polyacrylamide gel electrophoresis (BN-PAGE)**. Protein samples (~150 µg) were resuspended in 500 mM 6-aminohexanoic acid, 1 mM EDTA and 50 mM imidazole/HCl (pH 7.0) and solubilized with either Triton X-100 (Sigma), Digitonin (SERVA) or n-dodecyl-β-D-maltoside (DDM) (Sigma) using detergent:protein (w/w) ratios of 10:1, 6:1 and 3:1, respectively. The solubilized samples were centrifuged at $22,000 \times g$ for 20 min; 4 °C. The supernatants were recovered, supplemented with Coomassie-blue loading buffer and separated on either a 4–16% or 3–16% polyacrylamide gradient blue native gels as described previously[17]. For mass calibration, purified bovine heart mitochondria (50–100 µg) solubilized under the same conditions were run alongside each set of *Plasmodium* samples.

**Denaturing polyacrylamide gel electrophoresis (SDS-PAGE)**. Mature gametocyte or ABS parasite pellets were generated according to the procedure described above. Samples were lysed in SDS loading buffer (1% β-Mercaptoethanol, 0.004% Bromophenol blue, 6% glycerol, 2% SDS 50 mM Tris-HCl, pH 6.8) and heated for 5 min at 95 °C. Insoluble debris were pelleted by centrifugation ($20,000 \times g$, 5 min, RT). Supernatants were recovered and 20 µg protein was separated on SurePAGE Bis-Tris 4–12% gradient gel following the manufacturer's instructions (Genscript). A protein standards ladder (BioRad #1610374) was run alongside the samples for mass calibration.

**In-gel trypsin digestion**. After electrophoresis, the gels were fixed in 50% methanol, 10% acetic acid, 10 mM ammonium acetate, stained with 0.025% Coomassie-blue G-250 (SERVA) in 10% acetic acid for 30 min, destained in 10% acetic acid and kept in deionized water. A real-size colour picture was taken using an ImageScanner III (GE Healthcare) to prepare a template for the cutting procedure. The in-gel tryptic digestion was carried out following the method described in Heide et al.[18] with slight modifications. In brief, each gel lane was cut into 30 or 60 even slices starting at the bottom. Each slice was further diced into smaller pieces before being transferred to a filter microplate (96 wells, Millipore MABVN1250) prefilled with 200 µl 50% methanol, 50 mM ammonium hydrogen carbonate (AHC) per well. In order to remove the Coomassie dye, the gel pieces were incubated in the same solution at room temperature (RT) and washed by centrifugation ($1000 \times g$, short spin) until flow through was clear. For cysteine reduction, the gel pieces were incubated in 10 mM DL-dithiothreitol, 50 mM AHC for 60 min at RT under gentle agitation. The solution was removed by centrifugation at $1000 \times g$; short spin. In the next step, for cysteine alkylation, the gel pieces were incubated in 30 mM 2-chloroacetamide, 50 mM AHC for 45 min at RT and solution was removed as described above. The gel pieces were dehydrated in 50% methanol, 50 mM AHC for 15 min. The solution was removed and gel pieces were dried at RT for 45 min. Then, 20 µl of 5 ng/µl sequencing grade trypsin (Promega), 50 mM AHC, 1 mM CaCl2 were added to the dried gel pieces, incubated at 4 °C for 30 min before 50 µl more AHC were added and incubated overnight at 37 °C in a sealed bag. The next day, the peptides were collected in 96-well PCR microplates by centrifugation at $1000 \times g$; 15 s. The remaining peptides were eluted by incubating gel pieces with 50 µl 30% acetonitrile (ACN), 3% formic acid (FA) for 15 min at RT under gentle agitation and collected in the same PCR microplates. The peptide-containing solution was dried in a SpeedVac Concentrator Plus (Eppendorf) and the dried peptides were resuspended in 20 µl 5% ACN, 0.5% FA and stored at −20 °C for subsequent analysis.

**Mass spectrometry**. Resulting peptides were separated by liquid chromatography (LC) and analyzed by tandem mass spectrometry (MS/MS) in a Q-Exactive mass spectrometer equipped with an Easy nLC1000 nano-flow ultra-high-pressure LC system (Thermo Fisher Scientific). Briefly, peptides were separated using a 100 µm ID × 15 cm length PicoTip emitter column (New Objective) filled with ReproSil-Pur C18-AQ reverse-phase beads of 3 µm particle size and 120 Å pore size (Dr. Maisch GmbH) using linear gradients of 5–35% ACN, 0.1% FA (30 min), followed by 35–80% ACN, 0.1% FA (5 min) at a flow rate of 300 nl/min and a final column wash with 80% ACN (5 min) at 600 nl/min. The mass spectrometer was operated in positive mode switching automatically between MS and data-dependent MS/MS of the top 20 most abundant precursor ions. Full-scan MS mode (400–1400 $m/z$)

was set at a resolution of 70,000 $m/\Delta m$ with an automatic gain control target of $1 \times 10^6$ ions and a maximum injection time of 20 ms. Selected ions for MS/MS were analyzed using the following parameters: resolution 17,500 $m/\Delta m$, automatic gain control target $1 \times 10^5$; maximum injection time 50 ms; precursor isolation window 4.0 Th. Only precursor ions of charge $z = 2$ and $z = 3$ were selected for collision-induced dissociation. The normalized collision energy was set to 30% at a dynamic exclusion window of 60 s. A lock mass ion ($m/z = 445.12$) was used for internal calibration[91].

**Complexome profiling**. Raw MS data files from all slices were analyzed using MaxQuant (v1.5.0.25)[92]. For protein group identification peptide spectra were searched against a *P. falciparum* reference proteome (isolate 3D7, version March 21, 2020, obtained from uniprot.org) as well as a list of common contaminants; e.g., BSA and human keratins. Standard parameters were set for the searches, except for the following: N-term acetylation and methionine oxidation were allowed as variable modifications; up to two missed trypsin cleavages were allowed; cysteine carbamidomethylation as fixed modification; matching between runs was allowed and 2 min as matching time window; FDR as determined by target-decoy approach was set to 1%; 6 residues as minimal peptide length. To allow for abundance comparisons between samples, label-free quantification was applied to each detected protein in the form of iBAQ values. Potential differences in protein quantity and instrument sensitivity between runs were corrected by normalizing for the sum of total iBAQ values from each sample. Protein migration profiles were hierarchically clustered by an average linkage algorithm with centred Pearson correlation distance measures using Cluster 3.0[93]. Further analysis of the complexome profiles consisting of a list of proteins arranged depending on their similar migration patterns in the BN gel was performed in R[94] and the results were visualized using ggplot2[95]. The mass calibration was performed using the known masses of the mitochondrial OXPHOS complexes in bovine heart: CII (123 kDa); CIV (215 kDa); CIII (485 kDa); CV (700 kDa); CI (1000 kDa); supercomplex I-III ($S_0$, 1500 kDa); supercomplex I-III-IV ($S_1$, 1700 kDa); supercomplex I-III-IV$_2$ ($S_2$, 1900 kDa). Due to different migration properties in BN-PAGE, a separate mass calibration for soluble proteins was performed based on prior complexome profiling of bovine heart mitochondria solubilized in digitonin and soluble protein mass calibration curves described by Wittig et al.[64]. For the high molecular mass gametocyte profiles, one pooled sample was split into a 400 µg and a 200 µg lane. Since results were largely the same, data were averaged between these lanes for analysis. Individual profiles can be found in Supplementary Data 3.

**Homology detection**. In order to detect homologous proteins, we used profile-based sequence analysis tools. The sequence profile of each protein was queried between human, yeast, *Arabidopsis*, *Toxoplasma*, and *Plasmodium* proteomes and back, with the respective best hits, using HHpred[53]. Orthology was thus confirmed when retrieving the original query. Independently, these proteins were utilized to perform profile-based sequence analysis against the UniProtKB with Jackhmmer[96] in order to find orthologues among the species listed, examining always consistencies with aforementioned findings.

**Reporting summary**. Further information on experimental design is available in the Nature Research Reporting Summary linked to this paper.

## Data availability

All raw and processed complexome data generated in this study were deposited at the ComplexomE profiling DAta Resource (CEDAR) and can be retrieved under "CRX23 [www3.cmbi.umcn.nl/cedar/browse/experiments/CRX23]". Complete underlying datasets for Figs. 3, 5, 6, and 7 and further contextually relevant data are provided in Supplementary Data 1, 2, and 3. Source data are provided with this paper.

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

## Acknowledgements

We thank the molecular team members of the Malaria Research Group for fruitful discussions. We also thank Sergio Guerrero-Castillo for his support and helpful discussions. Furthermore, we would like to thank Joeri van Strien for helping in data exploration, helpful discussions and setting up CEDAR. F.E. and T.W.A.K. are supported by the Netherlands Organisation for Scientific Research (NWO-VIDI 864.13.009), A.C.O., U.B. and M.A.H. by the Netherlands Organization for Health Research and Development (TOP 91217009), U.B. by the Netherlands Organization for Scientific Research (TOP 714.017.004), and T.S.V. by the Swiss National Science Foundation (BSCGI0_157729). D.E. was funded by the PERISCOPE project of the Innovative Medicines Initiative 2 Joint Undertaking (No 115910), supported by the EFPIA, BMGF, and European Union's Horizon 2020 research and innovation programme.

## Author contributions

F.E. and A.C.O. performed experiments and analyzed results. M.K.L. performed transmission electron microscopy. D.M.E. and M.A.H. performed phylogenetic analysis and contributed illustrations. S.D.B and T.S.V provided the NF54/iGP2 line. U.B. provided conceptual advice and resources. F.E. prepared illustrations and wrote the first manuscript draft. T.W.A.K. conceived and designed the study, provided resources and edited the manuscript. All authors contributed to data interpretation and provided feedback on the manuscript. All authors approved the final version of the manuscript.

## Competing interests

The authors declare no competing interests.
