## [Peer Review File · Nature Communications]

REVIEWER COMMENTS

Reviewer #1 (Remarks to the Author):

In this manuscript, Evers and colleagues used complexome profiling in asexual blood stages and mature (stage V) gametocytes of *Plasmodium falciparum*. The authors use well-characterized protein complexes as a proof of concept for their approach, then proceed to identify the different components of the mitochondrial electron transport chain (ETC) complexes. Although this type of approach had already been implemented in a relative of *P. falciparum* (*Toxoplasma gondii*) this is the first time that the composition of the ETC complexes in *P. falciparum* is revealed in such an extent. The data sets generated in this paper will be of broad interest for anyone studying the apicomplexan mitochondria and its metabolic pathways. The authors also demonstrate that there is a 40-fold increase in the abundance of respiratory chain complex components in gametocytes compared to asexual blood stages and show that cristae are exclusively present in gametocytes. These observations confirm a long standing view in the field regarding the energetic and metabolic needs of the *P. falciparum* and set the scene for complexome profiling in the mosquito stages of this parasite. The only thing that I find disappointing is that some of the novel protein candidates discovered through complexome analysis were not preliminarily characterized.

Comments and critiques:

- The authors convincingly show the presence of cristae in the mitochondria of stage V gametocytes, but they do not mention anything about the earlier stages of gametogenesis. Did the authors look into those? TEM observations of other stages will likely determine when are the cristae starting to be present when could indicate when the parasites are shifting from glycolysis to OXPHOS.
- Line 168: "we termed RhopH associated protein 1 (RhopA1; 168 PF3D7_0220200; Fig. 2c)." I believe the authors forgot an H in RhopA1.
- Line 169: I think that localizing RhopHA1 by immunofluorescence will confirm that the protein is a novel RhopH component add to the evidence that the author's approach is working correctly.
- Table 1: maybe I missed it, but I do not see what is the meaning of the asterisk alongside the complex V ATP delta, epsilon, a and c subunits. Where those missing from the purifications?
- Figures 1e and 5b contain low-quality figures. Please upload their high resolution counterparts
- Figure 6: please indicate in the figure the corresponding respiratory chain complexes, otherwise it is a tad overwhelming.
- In the discussion, the authors mention that is challenging to estimate the biological function of the novel uncharacterized and apicomplexan-specific OXPHOS complex subunits. Did the authors try to identify conserved structures and/or domains in those proteins? If so they should mention it. Otherwise, a good starting point to assess those characteristics will be to use a protein fold recognition server or another approach to at least hypothesize about their possible function.

Reviewer #2 (Remarks to the Author):

The manuscript by Evers et al. entitled "Composition and stage dynamics of mitochondrial complexes in *Plasmodium falciparum*" describes the mitochondrial proteome organization of an important malaria parasite. Inspired by a combination of highly dynamic structural features and varying functions between the asexual blood stages (ABS) and the gametocyte stage, these organelles have previously been identified as a promising targets for anti-malarial drugs. Following confirmation of previous results on the ultrastructure of *P. falciparum* mitochondria by electron microscopy the authors use complexome profiling to evaluate the composition of their respiratory protein complexes. This technique has successfully been used to investigate respiratory

and non-respiratory protein complexes in mitochondrial samples from mammals, fungi, and plants by combining the sensitivity of MS-based protein detection with the separation resolution of blue native PAGE. In addition, the authors also compare the composition of *P. falciparum* respiratory complex with that of model organisms from different kingdoms and provide further insight into the proteomes of ABS and gametocyte cells.

The research presented in the manuscript is timely and important. The technical approaches are generally well suited for this type of research and data quality is high.

There are, however, some aspects which are in need of clarification:

1.) It is not clear, if the data presented in the paragraph entitled 'Validation with common eukaryotic Plasmodium-specific complexes' are produced from cellular extracts or from mitochondrial fractions, which are in the focus of this manuscript. In the latter case, the presence of these complexes must be due to co-purification with the mitochondrial fraction and I wonder if a cell extract would not have provided a better basis for validating the complexome profiling approach.

In any case, for a pure validation of the approach, the composition of cytosolic and ER complexes is discussed at too great a length. This distracts to some degree from original purpose of the manuscript, i.e. the investigation of the OXPHOS complexes (and supercomplexes) and confers a resource paper-character to the manuscript.

2.) The results on the *P. falciparum* respiratory complexes III and IV (and supercomplexes) are generally sound. They suggest unusually high molecular masses for respiratory complexes III and IV, which could be traced to the presence of additional, comigrating subunits. For some of these proteins, association with respiratory complexes was reported here for the first time. Unfortunately, despite its undisputed merits for protein complex research, complexome profiling is not suited to provide clear evidence for protein complex membership. This should be more openly discussed in the manuscript, also in respect to the newly identified, potential complex II subunits.

3.) The proposed stoichiometry of complex III and complex IV supercomplexes as shown in Figure 3C is disputable for the 'd' labeled supercomplex (CIII4 + CIV) judged by the relative average complex III and complex IV signal abundance of its neighboring supercomplexes. When compared to 'c', complex IV abundance should (in relation to complex III abundance) be decreased in supercomplex d. Instead, the opposite seems to be the case.

4.) In the 'Protein dynamics are in line with a significant metabolic shift in *P. falciparum* gametocytes' chapter it is mentioned that MS-analysis of SDS-separated cell extracts was performed excluding protein IDs which appear in slices representing a non-fitting molecular mass to exclude false positive IDs. This seems rather odd considering that exclusion of false positive IDs is usually performed based on peptide/protein scoring during database search. Please elaborate. Also, what is the advantage over a classical 'shotgun' approach?

5.) The supplemental material, mass calibration leaflet, shows diverging molecular masses for (I assume) membrane and soluble calibration. What has been used for the soluble calibration and why does it differ so profoundly from the BHM-calibration? What does Pred_sol_S0 (for example) mean?

Minor points:

Fig. 5C, figure legend: "Heat map showing SDHA and SDHA comigrating..." Should this read "...SDHA and SDHB comigrating..."?

L168, 169: Is RhopA1 the same as RhopHA1?

L417: as far as I am aware, abundance has no plural. Please check.

L1033: sentence stops rather abruptly.

Reviewer #3 (Remarks to the Author):

In the manuscript titled as "Composition and stage dynamics of mitochondrial complexes in *Plasmodium falciparum*", Evers et al., describe using the complexome profiling technique to dissect the subunit composition of mitochondrial respiratory complexes. The complexome profiling work is extensive and well done, and the results offer deeper insights in the composition and assembly of mitochondrial complexes. The study confirms previous findings and also uncovers new information on the divergent composition of the complexes. The study also demonstrates the differential abundance of the mitochondrial complexes by many fold in gametocyte stages, which again confirms previous findings that mitochondrial metabolism is required for the survival of gametocyte stage parasite. The paper is well written and the experiments are clearly described and easy to understand.

Specific points and questions on the manuscript, to be addressed by the authors in details, are given below.

The homooligomeric form of EXP2 protein appears to be between ~500 - ~600 kDa rather than ~800 kDa as stated (Line 212). In fig 2e, the arrow marking for ~630 kDa appears to be dislocated at ~530 kDa. The authors need to clarify this or correct it in case of a mistake.

The cytochrome C oxidase (CIV) complex from *Toxoplasma gondii* was recently characterised (Ref 10). This manuscript confirms that *Plasmodium* CIV includes the orthologs of many CIV subunits as reported in Ref 10. In addition, it is nice that this study also expands the annotation to correctly identify some of the subunits of apicomplexan CIV. It is also interesting that the NDUF4A has been identified as part of CIV complex here. Originally this protein was thought to be a component of the NADH Dehydrogenase complex-1 but recent findings have assigned it to CIV (Ref: Trends Endocrinol Metab. 2018 Jul;29(7):452-454 and references therein). In fact, this protein is found in many species lacking the mtETC complex I, thus strengthening its likely association with CIV complex. This study provides another line of evidence for this and this point can be addressed in the discussion section of the manuscript.

The authors also report the identification of the mitochondrial super complexes involving different stoichiometric associations between CIII and CIV complexes. These super complexes are thought to facilitate substrate channelling. However, given the markedly lower abundance of these supercomplexes, can the authors comment on their functional relevance in the parasite.

The data presented here for CV (mitochondrial ATP synthase) suggests that the ATP synthase has a dimer size of ~2.2 MDa. Authors also mention that the ~1 MDa dimer form reported in previous studies may actually be the monomer form (Line 345). I am not convinced by this conclusion as in both *P. falciparum* and *T. gondii*, it has been consistently shown that the dimer form of the protein is around ~1 MDa.

In case of *Plasmodium*, BN Page western blotting has clearly shown the complex to be present as a dimer at ~1.1 MDa and as monomer at ~550 kDa (Ref 59). Surprisingly, both of these forms are not detected in the experiments performed in this study. It must be noted that in the Ref 59 study, the Western blot shows a distinct band close to the well which probably is higher order (i.e., >1.1 MDa) complex. But this is in addition to the lower forms. As per data shown in Table-1, the molecular masses of subunits add up to ~1 MDa (with expected subunit counts). However, it is known that only the core subunits form the dimer and many of the novel subunits which presumably are accessory subunits are likely to be in single copy in the dimer form. My quick calculation showed that the core F1 subunits (alpha(3), beta(3), gamma(1), epsilon(1) delta(1)) and F0 subunits (a(1), b(1), c(10), d(1)), plus oscp (1) add up to a mass of ~800 kDa. A dimer of this would be ~1.6 MDa. This is very similar to the calculated mass of the core dimer in *T. gondii* (as in Ref 11 & 12) although in these studies the BN gels showed the dimer size around ~1 MDa as previously reported for *Plasmodium*. Other accessory subunits reported here add up to ~200 kDa and will give a total mass of ~1.8 MDa. This is based on the assumption that accessory subunits are at a stoichiometry of 1 per dimer, which may not be true. Data in Fig 5b shows that there is considerable signal in the ~1.8 MDa range as well, and in fact the novel subunit d-like protein is

more abundant at this mass range (gametocyte data).

Moreover, the fact that bulk of the F1-alpha, F1-beta and the OSCP subunits are in their monomeric form suggests that the CV is unstable in the experimental conditions used here. In previous studies in both Plasmodium and Toxoplasma such abundance of the monomeric forms of these proteins were not observed. In addition, the inability to detect the epsilon and delta subunits by complexome profiling (but detected in SDS PAGE profiling - Fig 6) strongly suggest that the F1 portion of the complex is disassociating under the experimental conditions. Most of the complex also appears to be failing to enter the resolving gel, which would suggest that there is aggregation issue. Thus, with respect to the CV, although it is good to have subunit composition confirmation from this study, I am not convinced regarding the migration and apparent size of the complex as reported here. The authors should clarify this and provide additional data, if necessary, to support their conclusions.

Line 127: ABS1Ma should be ABS1M (as per details from Sup. Table-1)

Line 288: It is stated that UQCRC is specifically lost from Apicomplexa, but it is shown in Fig 4A that it is absent in all alveolate species.

Line 291/292: Same point as above, COX7A & COX7C are absent in alveolate phylum.

Line 520: It is suggested that the lack of energetic constraints in the parasite might have allowed them to retain the bulky mitochondrial complexes. This assumption may not be correct, as it is known (and also discussed in this manuscript) that the mitochondrial energy metabolism is essential for growth and development of the parasite in the mosquito host. It is likely that there will be evolutionary constraints on the energy metabolism to facilitate efficient transition of the parasite through the mosquito to complete its life cycle.

Line 483: Rewrite, "It is tempting to speculate that ..." to "It is possible that..."

Line 627: "...gel pieces were dried at RT for 45 min at RT". "at RT" repeated twice, the second one can be removed.

Line 141: Reference 30 is for PlasmoDB. It is better to site the original reference for the proteomics data that is loaded in PlasmoDB. If it is unpublished data, the data depositors details as given in PlasmoDB can be given. Since there are many experiments loaded in PlasmoDB, it is not readily apparent which experiment is being referred to here.

Line 258: Fig.3b, peak a should be Fig.3c, peak a

Line 168 / 169 / 206: Is it RhopA1 or RhopHA1. Need to be consistent;

Figure-1: The magnification scale for the insets (zoomed in areas) of the electron micrographs can be added.

Supp. Fig-1: The same micrograph is shown twice (flipped horizontally). Not sure it needs to be shown twice, especially since the magnification appears to be the same. One of the micrographs (maybe the bottom one) can be omitted.

Ref 57 is same as 11, and so can be removed. Ref 57 to be replace with Ref 11 in line 331.

The legend for Supplementary figure 6 appears to be incomplete. Line 1033: correct the spelling for 'position'.

The authors claim that the increased size of the mitochondrial complexes native size predates origin of Apicomplexa (Line 511). Did the authors find evidence for the acquisition of these large complexes by apicomplexa from ancestral species? Moreover, it is also pointed out that the size increase is particular for Plasmodium falciparum (Line 509). What about other Plasmodium species? Have the authors checked RNA data to make sure that the predicted gene structure is

correct for proteins with extra sequences. This can verify the unusually large size of some of the canonical components of the mitochondrial complexes.

In discussion (line 503 - 509), the authors suggest the parasite specific increase in size of the complexes. At least for CV, as point out above, the migration pattern and size appear not to be consistent with previous findings. The data needs to be verified further before making conclusions on CV.

Summary

First of all, we would like to thank the reviewers for the positive response and very much appreciate the recognition of both the good technical execution of this study and relevance to the scientific community. We also like to thank you for the valuable comments and suggestions, which have further strengthened our manuscript. Before moving towards the detailed point-by-point responses, we first like to highlight a two major points here.

Reviewer #2 and #3 highlighted limitations to our suggestions of potentially higher oligomeric states for ATP synthase as well as diverging mass assignments from prior studies. In order to address these issues we included an additional experiment in which we resolve gametocyte mitochondrial fractions on a 3-16% BN-PAGE instead of the 4-16% gradient that was utilized for all other samples. This adjustment allows the higher oligomeric state to enter the gel, while precipitated protein, as suggested by reviewer #2, would still not enter the gel. Doing so we provide tentative evidence for the presence of higher oligomeric states, identify additional subunits and strengthen our previous observation of an unusually large dimer. We also use this data to corroborate our respiratory supercomplex observations, adding a biological replicate with an altered sample separation technique.

During revision of this manuscript, another beautiful study, titled "ATP synthase hexamer assemblies shape cristae of *Toxoplasma mitochondria*" (<https://doi.org/10.1038/s41467-020-20381-z>) has been published in *Nature Communications* that adds novel insights that are highly relevant in the context of this study. The study provides clear evidence of presence of an unexpectedly large observed ATP synthase dimer, as well as confirming the presence of higher state ATP synthase oligomers through cryoEM in *T. gondii*, both of which support hypotheses brought forward in this study. While data is only derived from a related species and not the same model, it caters the criticism raised by reviewer #2 that our data is in conflict with earlier publications, suggesting a much smaller ATP synthase dimer, and counters the suggestion that the intensity in the stacking gel represents precipitated protein. As this publication represents the most definitive investigation on ATP synthase in Apicomplexa yet, proposed nomenclature for novel subunits was also adopted for this manuscript.

Please find the detailed point-by-point response below. Significant textual changes in the manuscript are highlighted in blue and figures were altered to include the new data and where applicable to replace lower resolution panels with high-resolution counterparts.

REVIEWER COMMENTS

Reviewer #1 (Remarks to the Author):

In this manuscript, Evers and colleagues used complexome profiling in asexual blood stages and mature (stage V) gametocytes of *Plasmodium falciparum*. The authors use well-characterized protein complexes as a proof of concept for their approach, then proceed to identify the different components of the mitochondrial electron transport chain (ETC) complexes. Although this type of approach had already been implemented in a relative of *P. falciparum* (*Toxoplasma gondii*) this is the first time that the composition of the ETC complexes in *P. falciparum* is revealed in such an extent. The data sets generated in this paper will be of broad interest for anyone studying the apicomplexan mitochondria and its metabolic pathways. The authors also demonstrate that there is a 40-fold

increase in the abundance of respiratory chain complex components in gametocytes compared to asexual blood stages and show that cristae are exclusively present in gametocytes. These observations confirm a long standing view in the field regarding the energetic and metabolic needs of the *P. falciparum* and set the scene for complexome profiling in the mosquito stages of this parasite. The only thing that I find disappointing is that some of the novel protein candidates discovered through complexome analysis were not preliminary characterized.

We thank the reviewer for the positive appraisal of our study and recognition of the significance of the data.

Reviewer #1 suggests that complexome profiling has already been implemented in *T. gondii* to reveal similar features of the respiratory chain. This statement is presumably based on the preprint by Maclean *et al.* 2020. While it was accessible to the public one month prior to this study, it is still not peer-reviewed work and does not predate this work by a significant margin. Additionally, first public appearance of complexome profiling of mitochondrial complexes in Apicomplexa was a poster presentation containing a subset of this work at the Molecular Parasitology Meeting 2019 in Woods Hole, significantly predating any public appearance of the *Toxoplasma* work to the public. The first peer-reviewed appearance of complexome profiling as an approach has been demonstrated in *P. falciparum* by Hillier *et al.* in 2019 (<https://doi.org/10.1016/j.celrep.2019.07.019>) but presumably due to high sample complexity or stage constraints, was unable to identify any assembled mtETC complexes.

We appreciate and share the eagerness of reviewer #1 to learn more about the newly identified complex members and naturally we are in the process of exploring the roles of some of these. However, instead of including limited preliminary and inconclusive data at this stage, we decided to disseminate these at a later when the work is more complete. This also left more room for a more comprehensive presentation doing justice to the wealth of data presented in this study.

Comments and critiques:

- The authors convincingly show the presence of cristae in the mitochondria of stage V gametocytes, but they do not mention anything about the earlier stages of gametogenesis. Did the authors look into those? TEM observations of other stages will likely determine when are the cristae starting to be present when could indicate when the parasites are shifting from glycolysis to OXPHOS.

The manuscript is focused on the direct comparison of asexual and mature sexual blood stages to ensure a maximum differentiation in both microscopy and proteomic approaches thus providing the most meaningful data to inform our future work. Again, we share a keen interest in the timeline of cristae biogenesis and how this unique process is reflected on an ultrastructural and proteomic level. The process of cristae biogenesis throughout gametocytogenesis is one of the main foci of the lab, but given that these kinds of studies are rather time and resource intensive, these ongoing efforts fall well beyond the scope of this study.

- Line 168: “we termed RhopH associated protein 1 (RhopA1; 168 PF3D7_0220200; Fig. 2c).” I believe the authors forgot an H in RhopA1.

Changed accordingly

- Line 169: I think that localizing RhopHA1 by immunofluorescence will confirm that the protein is a novel RhopH component add to the evidence that the author’s approach is working correctly.

We agree that this is an interesting novel complex candidate to study further but a straightforward immunofluorescence experiment will not be sufficient to confirm membership of the RhopH complex. A more convincing, but also much more time-consuming approach would be to perform co-immunoprecipitation of the tagged component, which would require months of additional work on something that in the end is only used to validate and exemplify the approach used for the main findings of this paper. Other labs are much better positioned (and interested) to confirm this new RhopH complex member and its role. Furthermore, expanding on this part of the story would distract from the main focus of the paper on the comparison between asexual and sexual blood stages and the mtETC complexes, which is also in direct conflict with the request of reviewer #2 to reduce attention paid to the validation.

- Table 1: maybe I missed it, but I do not see what is the meaning of the asterisk alongside the complex V ATP delta, epsilon, a and c subunits. Where those missing from the purifications?

Yes, those were not found by MS but still included in the table due to their high sequence conservation to the canonical components and essentiality to the structure and functioning of complex V. Text was changed in order to make this clearer to the reader and a footnote has been added to the table. Additional subunits were identified with the addition of the new HMM profiles, leaving only ATP α and ATP γ as canonical subunits that remain unidentified.

- Figures 1e and 5b contain low-quality figures. Please upload their high resolution counterparts

Reviewer #1 presumably referred to figure 2e as it contains a panel with lower resolution than the other ones and figure 1e does not exist. High-resolution images are now provided for all figures.

- Figure 6: please indicate in the figure the corresponding respiratory chain complexes, otherwise it is a tad overwhelming.

Changed accordingly

- In the discussion, the authors mention that is challenging to estimate the biological function of the novel uncharacterized and apicomplexan-specific OXPHOS complex subunits. Did the authors try to identify conserved structures and/or domains in those proteins? If so they should mention it. Otherwise, a good starting point to assess those characteristics will be to use a protein fold recognition server or another approach to at least hypothesize about their possible function.

As suggested we have expanded the discussion to include the following section on potential roles of the novel subunits based on computational support for the protein structures: „For the novel uncharacterized and apicomplexan-specific OXPHOS complex subunits, it is challenging to estimate biological significance or function. With the exception of ApiCOX13, of which the human orthologue binds iron-sulphur clusters, and the addition of a glycosyl transfer domain to PfCOX6C the new CIII and CIV subunits do not have detectable homologues outside the Myzozoa or indeed any recognizable functional domains that could give us a hint about their potential function. A large proportion of them have predicted transmembrane helices, specifically all three new CIII subunits and nine of the twelve new CIV subunits (Supplementary Table 3). We can speculate they surround CIII and CIV in the membrane plane as has been observed before for supernumerary OXPHOS subunits in opisthokonts (55). Two of the new subunits might even be functional and structural replacements of the two missing subunits of CIV. However, as we have observed in the evolution of CI in Metazoa, evolutionary new subunits do not necessarily occupy the same location in the complex where subunits are missing (79).“

Supplementary table 3, is a newly added table containing probability plots for transmembrane helices and inside and outside positioning of all AA residues of each putative complex III/IV protein in *P. falciparum* as well as their orthologues in *S. cerevisiae* and *H. sapiens* as generated by the TMHMM Server v. 2.0.

Reviewer #2 (Remarks to the Author):

The manuscript by Evers et al. entitled “Composition and stage dynamics of mitochondrial complexes in *Plasmodium falciparum*” describes the mitochondrial proteome organization of an important malaria parasite. Inspired by a combination of highly dynamic structural features and varying functions between the asexual blood stages (ABS) and the gametocyte stage, these organelles have previously been identified as a promising targets for anti-malarial drugs.

Following confirmation of previous results on the ultrastructure of *P. falciparum* mitochondria by electron microscopy the authors use complexome profiling to evaluate the composition of their respiratory protein complexes. This technique has successfully been used to investigate respiratory and non-respiratory protein complexes in mitochondrial samples from mammals, fungi, and plants by combining the sensitivity of MS-based protein detection with the separation resolution of blue native PAGE. In addition, the authors also compare the composition of *P. falciparum* respiratory complex with that of model organisms from different kingdoms and provide further insight into the proteomes of ABS and gametocyte cells.

The research presented in the manuscript is timely and important. The technical approaches are generally well suited for this type of research and data quality is high.

We thank the reviewer for the positive appraisal of our study both in its significance and execution.

There are, however, some aspects which are in need of clarification:

1.) It is not clear, if the data presented in the paragraph entitled ‘Validation with common eukaryotic *Plasmodium*-specific complexes’ are produced from cellular extracts or from mitochondrial fractions, which are in the focus of this manuscript. In the latter case, the presence of these complexes must be due to co-purification with the mitochondrial fraction and I wonder if a cell extract would not have provided a better basis for validating the complexome profiling approach.

In any case, for a pure validation of the approach, the composition of cytosolic and ER complexes is discussed at too great a length. This distracts to some degree from original purpose of the manuscript, i.e. the investigation of the OXPHOS complexes (and supercomplexes) and confers a resource paper-character to the manuscript.

We agree that we spend a significant amount of time discussing non-mitochondrial complexes, which can distract from the overall OXPHOS focus of this study. Nevertheless, due to the relative novelty of this approach in Apicomplexa, we feel the need to validate the approach by reconstituting known parasite complexes. To encourage malaria researchers with an interest in other protein complexes to embrace this exciting method, we tried to choose complexes from which we can derive some degree of novel insight that is of value to the general parasitology research community. While we could reduce length of this section, we feel that this is in conflict with reviewer #1 that asked us to expand on parts of this section, reassuring us that we have found the right balance.

In lines 132-135 preceding the paragraph it reads: “It should be noted that abundant proteins from other *P. falciparum* cell compartments were also readily identified. Taking advantage of the latter and to validate the approach, we first investigated whether well-known and previously described complexes could be identified correctly and what the impact was of different isolation methods (Fig. 2).” which should make it clear to the reader that data shown in Fig.2 are also derived from mitochondrial fractions. To make evident from looking at the figure alone, we added “Common eukaryotic and parasite-specific protein complexes identified in mitochondrially enriched fractions of *P. falciparum*” to the figure legend of Fig. 2.

2.) The results on the *P. falciparum* respiratory complexes III and IV (and supercomplexes) are generally sound. They suggest unusually high molecular masses for respiratory complexes III and IV, which could be traced to the presence of additional, comigrating subunits. For some of these proteins, association with respiratory complexes was reported here for the first time. Unfortunately, despite its undisputed merits for protein complex research, complexome profiling is not suited to provide clear evidence for protein complex membership. This should be more openly discussed in the manuscript, also in respect to the newly identified, potential complex II subunits.

The following sentence was added to the “Succinate dehydrogenase – respiratory chain complex II” section to reflect the lack of direct evidence: “Additionally, as complexome profiling provides no direct evidence of interaction and there are no other reports or studies from this or other systems that could cross-validate these putative novel subunits, further studies will be needed to verify our findings. “

3.) The proposed stoichiometry of complex III and complex IV supercomplexes as shown in Figure 3C is disputable for the ‘d’ labeled supercomplex (CIII4 + CIV) judged by the relative average complex III and complex IV signal abundance of its neighboring supercomplexes. When compared to ‘c’, complex IV abundance should (in relation to complex III abundance) be decreased in supercomplex d. Instead, the opposite seems to be the case.

We agree and have changed the assignment of supercomplexes in Fig. 3 accordingly. Additionally, we performed an additional experiment analyzing a biological replicate of the GCT3D conditions but instead separated on a 3-16% acrylamide gradient and only analyzing the 1-7MDa range to save machine and processing time (Supp. Fig. 5). This biological replicate under different separation conditions lends additional credence to the larger assemblies that could be identified.

4.) In the ‘Protein dynamics are in line with a significant metabolic shift in *P. falciparum* gametocytes’ chapter it is mentioned that MS-analysis of SDS-separated cell extracts was performed excluding protein IDs which appear in slices representing a non-fitting molecular mass to exclude false positive IDs. This seems rather odd considering that exclusion of false positive IDs is usually performed based on peptide/protein scoring during database search. Please elaborate. Also, what is the advantage over a classical ‘shotgun’ approach?

The removal of false positive IDs was also done during database search following the usual criteria for the analysis of MS data. However, by providing information on the apparent mass of proteins prior gel electrophoresis adds another criterion that allowed us to exclude false positives from the dataset. If peptides matched to a protein group migrate at markedly different form the expected molecular mass or even at an apparent mass deviating from that of other peptides of the same protein group, it is sensible to suggest that they are false positives. If this removal of non-matching peptides was not performed, results did not change by a big margin. The only exception is ApiCOX18,

which is markedly more abundant in ABS due to a supposedly unique ApiCOX18 peptide migrating at ~290 kDa apparent molecular mass, despite only belonging to a 16kDa protein under reducing and denaturing conditions.

5.) The supplemental material, mass calibration leaflet, shows diverging molecular masses for (I assume) membrane and soluble calibration. What has been used for the soluble calibration and why does it differ so profoundly from the BHM-calibration? What does Pred_sol_S0 (for example) mean?

Soluble and membrane proteins interact differently with detergents and membrane proteins also have different migration properties due to their association with lipids. This necessitates separate mass calibration. This concept is explored in a study by Wittig et al. ([10.1074/mcp.M900526-MCP200](https://doi.org/10.1074/mcp.M900526-MCP200)). The BHM calibration for membrane proteins is done by observing the visible bands and matching them to the slices that the samples were cut in. For the soluble proteins, not sufficient distinct bands are visible by Coomassie staining. However, BHM were analyzed by complexome profiling before and mass calibration of soluble proteins relative to membrane proteins was performed on that basis. As these soluble masses at positions in the gel are only implied by the position of the membrane complexes they are assigned as (Pred_Sol). To make this more clear, the method section was extended with a description of soluble mass calibration and the following additional consideration was added to the discussion:

“While it has been shown that migration in BN-PAGE is also influenced by factors other than mass such as ability to bind Coomassie dye, lipid association and intrinsic negative charges (62), our mass estimations do match expected mass with composition observed in this study. Additionally, mass estimations are based on OXPHOS complexes in BHM, which should be representative and have comparable electrophoretic properties to OXPHOS complexes in *P. falciparum*. Conversely this suggest that for complexome profiling of other compartments, a different set of proteins might be more representative of general electrophoretic properties.”

Minor points:

Fig. 5C, figure legend: “Heat map showing SDHA and SDHA comigrating...” Should this read “...SDHA and SDHB comigrating...”?

Corrected accordingly

L168, 169: Is RhopA1 the same as RhopHA1?

Yes, this an oversight on our part and was fixed.

L417: as far as I am aware, abundance has no plural. Please check.

Corrected accordingly

L1033: sentence stops rather abruptly.

Corrected accordingly

Reviewer #3 (Remarks to the Author):

In the manuscript titled as “Composition and stage dynamics of mitochondrial complexes in *Plasmodium falciparum*”, Evers et al., describe using the complexome profiling technique to dissect

the subunit composition of mitochondrial respiratory complexes. The complex some profiling work is extensive and well done, and the results offer deeper insights in the composition and assembly of mitochondrial complexes. The study confirms previous findings and also uncovers new information on the divergent composition of the complexes. The study also demonstrates the differential abundance of the mitochondrial complexes by many fold in gametocyte stages, which again confirms previous findings that mitochondrial metabolism is required for the survival of gametocyte stage parasite. The paper is well written and the experiments are clearly described and easy to understand.

We thank the reviewer for the positive appraisal of our study.

Specific points and questions on the manuscript, to be addressed by the authors in details, are given below.

The homooligomeric form of EXP2 protein appears to be between ~500 - ~600 kDa rather than ~800 kDa as stated (Line 212). In fig 2e, the arrow marking for ~630 kDa appears to be dislocated at ~530 kDa. The authors need to clarify this or correct it in case of a mistake.

We apologize and thank reviewer 2 for pointing out the inconsistencies between text and figure, which were corrected accordingly.

The cytochrome C oxidase (CIV) complex from *Toxoplasma gondii* was recently characterised (Ref 10). This manuscript confirms that *Plasmodium* CIV includes the orthologs of many CIV subunits as reported in Ref 10. In addition, it is nice that this study also expands the annotation to correctly identify some of the subunits of apicomplexan CIV. It is also interesting that the NDUFA4 has been identified as part of CIV complex here. Originally this protein was thought to be a component of the NADH Dehydrogenase complex-1 but recent findings have assigned it to CIV (Ref: Trends Endocrinol Metab. 2018 Jul;29(7):452-454 and references therein). In fact, this protein is found in many species lacking the mtETC complex I, thus strengthening its likely association with CIV complex. This study provides another line of evidence for this and this point can be addressed in the discussion section of the manuscript.

We thank the reviewer for bringing our attention to this interesting consistent behavior of NDUFA4. The following sentence was added to the results section to reflect original wrong association with complex I “NDUFA4 was originally identified as a complex I subunit but has later been shown to be a stoichiometric complex IV component (51), which is consistent with our observations “

The authors also report the identification of the mitochondrial super complexes involving different stoichiometric associations between CIII and CIV complexes. These super complexes are thought to facilitate substrate channelling. However, given the markedly lower abundance of these supercomplexes, can the authors comment on their functional relevance in the parasite.

We share this interest in the functional relevance of mitochondrial supercomplexes. However, at this point even in much more well understood systems their exact role or relevance is still a debated, which makes it challenging to speculate on their role in *Plasmodium* biology. We have added the following section to the discussion: “At this point, it is challenging to suggest a role the putative respiratory supercomplexes might play in *Plasmodium* biology. In mammalian systems, it has been suggested that supercomplexes play an important role in reduction of mitochondrial reactive oxygen species formation and to facilitate more efficient substrate channelling. They have also been suggested to play a role in preventing random

protein aggregation in the protein rich inner mitochondrial membrane. However, all these suggested roles have been challenged and no clear consensus has emerged (86). While our data suggest that the free form respiratory complexes are the dominant species in *P. falciparum* mitochondria, it is hard to estimate whether this balance is representative of the *in vivo* abundances and what impact solubilization and electrophoresis procedures have on stability of the supercomplexes. Though the low detection levels in ABS parasites do not allow any firm conclusions, the apparent abundance of supercomplexes in gametocytes could hint at their importance for a more efficient functioning. One conclusion that can be drawn though, is that cristae are not a prerequisite of supercomplex formation, though conversely OXPHOS supercomplexes may be important for forming or maintaining cristae.“

The data presented here for CV (mitochondrial ATP synthase) suggests that the ATP synthase has a dimer size of ~2.2 MDa. Authors also mention that the ~1 MDa dimer form reported in previous studies may actually be the monomer form (Line 345). I am not convinced by this conclusion as in both *P. falciparum* and *T. gondii*, it has been consistently shown that the dimer form of the protein is around ~1 MDa.

In case of Plasmodium, BN Page western blotting has clearly shown the complex to be present as a dimer at ~1.1 MDa and as monomer at ~550 kDa (Ref 59). Surprisingly, both of these forms are not detected in the experiments performed in this study. It must be noted that in the Ref 59 study, the Western blot shows a distinct band close to the well which probably is higher order (i.e., >1.1 MDa) complex. But this is in addition to the lower forms. As per data shown in Table-1, the molecular masses of subunits add up to ~1 MDa (with expected subunit counts). However, it is known that only the core subunits form the dimer and many of the novel subunits which presumably are accessories subunits are likely to be in single copy in the dimer form. My quick calculation showed that the core F1 subunits (alpha(3), beta(3), gamma(1), epsilon(1) delta(1)) and F0 subunits (a(1), b(1), c(10), d(1)), plus oscp (1) add up to a mass of ~800 kDa. A dimer of this would be ~1.6 MDa. This is very similar to the calculated mass of the core dimer in *T. gondii* (as in Ref 11 & 12) although in these studies the BN gels showed the dimer size around ~1 MDa as previously reported for Plasmodium. Other accessory subunits reported here add up to ~200 kDa and will give a total mass of ~1.8 mDa. This is based on the assumption that accessory subunits are at a stoichiometry of 1 per dimer, which may not be true. Data in Fig 5b shows that there is considerable signal in the ~1.8 MDa range as well, and in fact the novel subunit d-like protein is more abundant at this mass range (gametocyte data).

Moreover, the fact that bulk of the F1-alpha, F1-beta and the OSCP subunits are in their monomeric form suggests that the CV is unstable in the experimental conditions used here. In previous studies in both Plasmodium and Toxoplasma such abundance of the monomeric forms of these proteins were not observed. In addition, the inability to detect the epsilon and delta subunits by complexesome profiling (but detected in SDS PAGE profiling - Fig 6) strongly suggest that the F1 portion of the complex is disassociating under the experimental conditions. Most of the complex also appears to be failing to enter the resolving gel, which would suggest that there is aggregation issue. Thus, with respect to the CV, although it is good to have subunit composition confirmation from this study, I am not convinced regarding the migration and apparent size of the complex as reported here. The authors should clarify this and provide additional data, if necessary, to support their conclusions.

We appreciate the possibility that the parasite-specific subunits do not behave as stoichiometric subunits. Furthermore, while we do not see all the subunits that were observed, we still think that it is highly unlikely that the dimer is 1 MDa, due to similar calculations reviewer #2 has put forward. An

explanation for the diverging results could be that mass calibration in prior studies was based on commercial markers, which are made of soluble proteins that have different electrophoretic properties in BN-PAGE compared to membrane proteins. In this study, we estimate mass based on migration of respiratory complexes in bovine heart mitochondria, which should be much more representative of the electrophoretic properties of our protein complexes of interest. We have added Supplementary Figure 9 to the manuscript which shows poor performance of NativeMARK (the most commonly used commercial marker from Invitrogen) compared to bovine heart mitochondria and divergence of the resulting mass calibration curve, which can explain discrepancy between studies.

To further address reviewer #3 concerns, we made a biological replicate of the G3CD sample and instead separated it on a 3-16% gradient, allowing complexes of up to 10 MDa enter the gel. This confirmed our previous results and showed consistent comigration at an apparent mass of 2.2 MDa, while also showing comigration at larger apparent masses, supportive of the hypothesis of higher state CV oligomers.

During the preparation of the rebuttal, it has also been shown that at least in *T. gondii*, the novel subunits for the most part behave stoichiometrically, the c-ring does indeed contain 10 copies and that our mass suggestions roughly match the composition that has been observed by cryoEM (<https://doi.org/10.1038/s41467-020-20381-z>) Additionally, the authors have found that ATP synthase associated into hexamers, which is supportive of the thesis of higher state oligomers of ATP synthase being stuck at the stacking gel interface.

Line 127: ABS1Ma should be ABS1M (as per details from Sup. Table-1)
This sample is now termed ABS1Ma in both text and table.

Line 288: It is stated that UQCRC is specifically lost from Apicomplexa, but it is shown in Fig 4A that it is absent in all alveolate species.

Corrected accordingly

Line 291/292: Same point as above, COX7A & COX7C are absent in alveolate phylum.

Corrected accordingly

Line 520: It is suggested that the lack of energetic constraints in the parasite might have allowed them to retain the bulky mitochondrial complexes. This assumption may not be correct, as it is known (and also discussed in this manuscript) that the mitochondrial energy metabolism is essential for growth and development of the parasite in the mosquito host. It is likely that there will be evolutionary constraints on the energy metabolism to facilitate efficient transition of the parasite through the mosquito to complete its life cycle.

We thank the reviewer for this objection. We have added the following sentence to the corresponding section in the discussion: „This theory however, is not entirely compatible with the observation that colonization of the mosquito host requires efficient respiration and represents a major bottleneck in the *Plasmodium* life cycle (6).“

Line 483: Rewrite, “It is tempting to speculate that ...” to “It is possible that....”

Corrected accordingly

Line 627: “..gel pieces were dried at RT for 45 min at RT”. “at RT” repeated twice, the second one can be removed.

Corrected accordingly

Line 141: Reference 30 is for PlasmoDB. It is better to site the original reference for the proteomics data that is loaded in PlasmoDB. If it is unpublished data, the data depositors details as given in PlasmoDB can be given. Since there are many experiments loaded in PlasmoDB, it is not readily apparent which experiment is being referred to here.

We do agree that in most cases a blanket reference to PlasmoDB is not very helpful. In this case, the intention was to refer to the complete lack of MS detection in any experiment that has been loaded onto PlasmoDB. The alternative would be to cite all relevant proteomic studies that are shown on PlasmoDB which would require us to use ~8 references to make this relatively minor point.

To make this clearer, we have changed the sentence to:

“All previous proteomics experiments for which data are available on PlasmoDB (<https://plasmodb.org>) (30) also failed to detect EMC6 in ABS parasites and gametocytes despite it having similar transcription profiles as other EMC components, indicating challenging detection by MS or absence.”

Line 258: Fig.3b, peak a should be Fig.3c, peak a

Corrected accordingly

Line 168 / 169 / 206: Is it RhopA1 or RhopHA1. Need to be consistent;

RhopHA1 is correct and was fixed accordingly

Figure-1: The magnification scale for the insets (zoomed in areas) of the electron micrographs can be added.

Scale bar has been added to zoomed areas

Supp. Fig-1: The same micrograph is shown twice (flipped horizontally). Not sure it needs to be shown twice, especially since the magnification appears to be the same. One of the micrographs (maybe the bottom one) can be omitted.

Bottom micrograph was omitted

Ref 57 is same as 11, and so can be removed. Ref 57 to be replace with Ref 11 in line 331.

Corrected accordingly

The legend for Supplementary figure 6 appears to be incomplete. Line 1033: correct the spelling for ‘position’.

Corrected accordingly

The authors claim that the increased size of the mitochondrial complexes native size predates origin of Apicomplexa (Line 511). Did the authors find evidence for the acquisition of these large complexes by apicomplexa from ancestral species? Moreover, it is also pointed out that the size increase is

particular for *Plasmodium falciparum* (Line 509). What about other *Plasmodium* species? Have the authors checked RNA data to make sure that the predicted gene structure is correct for proteins with extra sequences. This can verify the unusually large size of some of the canonical components of the mitochondrial complexes.

We share the reviewer's interest in the apparent size increases of OXPHOS complexes. We did discuss likely origin of the size increases in the discussion: "While enlarged OXPHOS complexes have been observed before (54), the size increases in *P. falciparum* are remarkably large. These increases occurred roughly at the time of the origin of multicellular clades, *i.e.* before the origin of the Myzozoa ~1,300 million years ago, or considerably later in evolution before the origin of the Apicomplexa ~900 million years ago (82). "

While we did not do specifically discuss other *Plasmodium* species, the additional subunits appear to all have orthologues across all *Plasmodium* species and complex composition is still largely conserved even in *Toxoplasma* species, suggesting that these complex compositions are likely conserved in other *Plasmodium* species too.

In order to check the gene models of the novel subunits, we did look at transcript and proteomic evidence from other studies as well as this study, which both seem to confirm the gene model in the current genome annotation. The exception for this is PF3D7_0809250, which is not included as a gene in the latest PlasmoDB build (but is on uniprot) but its orthologues in *Toxoplasma* spp are annotated and have RNA and protein level evidence. A PDF file showing this is available as part of the rebuttal but not included as a supplementary figure.

In discussion (line 503 - 509), the authors suggest the parasite specific increase in size of the complexes. At least for CV, as point out above, the migration pattern and size appear not to be consistent with previous findings. The data needs to be verified further before making conclusions on CV.

See comment on earlier point about CV

REVIEWERS' COMMENTS

Reviewer #1 (Remarks to the Author):

I believe that the authors were able to address all the concerns that I raised and did an excellent job addressing the concerns of the other reviewers, particularly regarding the apparent size of CV. The authors should also mention the now recently published (it was a pre-print when the authors first submitted their manuscript) complexome profile of *T. gondii* mitochondria, where mass calibration suggests that the CV dimer in *T. gondii* has a total mass of ~1860 kDa, a value in range to the author's estimation of the molecular weight of CV dimers (and monomers) in *P. falciparum*.

Some minor comments:

-Line 328-329: I do not think the parentheses in "(predicted)" are necessary

-Figure 6b legend: "Putative components of ATP synthase detected in complexome profiles (upper section) or only detected in SDS profiles but identified as an ATP synthase component in *T. gondii* (11, 58) (lower section). I do not see any clear distinction between the upper and the lower sections of the figure. Please make a clear distinction

Reviewer #2 (Remarks to the Author):

The authors have responded to my critique/comments in a very convincing manner and I would like to congratulate them to their truly interesting manuscript.

Reviewer #3 (Remarks to the Author):

In the revised version of the manuscript titled as "Composition and stage dynamics of mitochondrial complexes in *Plasmodium falciparum*", Evers et al., have addressed queries from this reviewer satisfactorily.

Especially with regards to the oligomeric status of CV, the authors have now carried out extensive native mass calibration studies with commercial native markers as well as with bovine heart mitochondria, to demonstrate that the native complexes from mitochondria do not match in electrophoretic mobility with the calibration markers. I agree that the detergent extracted parasite mitochondrial complexes compare well to the detergent treated BHM extracts, and this likely provides an excellent way to determine the native mass more accurately. Thus the mass of CV dimer appears to be ~2.2 Mda. However, it is also evident that the presence of detergent and type of detergent greatly affects the mobility of native proteins (based on the BHM data). In this context, I am wondering whether the NativeMARK proteins will migrate differently in the presence of detergent as well. Nevertheless BHM based calibration probably more accurately represents the true mass of *Plasmodium* CV. Further, this is also in agreement with the formation of higher order oligomeric CV complex that has now been conclusively demonstrated in the related apicomplexan parasite *Toxoplasma gondii*.

The fact that the function for many of the novel proteins associated with CIII, CIV and CV are yet to be deciphered provided rich scope for further experimental work with these proteins & complexes; from structural and biochemical perspective. Moreover the presence of orthologs for many of these novel proteins in other apicomplexan and myxozoan species indicates conserved function and provides the opportunity to further study the evolutionary acquisition and retention of these proteins.

Minor Correction:

Table-I: for CV complex, the calculated monomer / dimer mass value is aligned to column-2 instead of column-3.

The manuscript is now significantly updated with new information, which clarifies many of the original queries raised. I now recommend accepting the manuscript for publication.

Reviewers' Comments – Point-by-point response

Reviewer #1 (Remarks to the Author):

I believe that the authors were able to address all the concerns that I raised and did an excellent job addressing the concerns of the other reviewers, particularly regarding the apparent size of CV. The authors should also mention the now recently published (it was a pre-print when the authors first submitted their manuscript) complexome profile of *T. gondii* mitochondria, where mass calibration suggests that the CV dimer in *T. gondii* has a total mass of ~1860 kDa, a value in range to the author's estimation of the molecular weight of CV dimers (and monomers) in *P. falciparum*.

We thank the reviewer for positive appraisal of our study and the kind words. We have now also included a reference to the publication indicated by the reviewer in lines 318 and 356.

Some minor comments:

-Line 328-329: I do not think the parentheses in “(predicted)” are necessary

Changed accordingly

-Figure 6b legend: “Putative components of ATP synthase detected in complexome profiles (upper section) or only detected in SDS profiles but identified as an ATP synthase component in *T. gondii* (11, 58) (lower section). I do not see any clear distinction between the upper and the lower sections of the figure. Please make a clear distinction

This was a leftover description from a previous iteration of the figure that no longer reflects the contents of the figure. The figure legend has now been adjusted to reflect the actual content of Figure 6b.

Reviewer #2 (Remarks to the Author):

The authors have responded to my critique/comments in a very convincing manner and I would like to congratulate them to their truly interesting manuscript.

We thank the reviewer for positive appraisal of our study and the kind words.

Reviewer #3 (Remarks to the Author):

In the revised version of the manuscript titled as “Composition and stage dynamics of mitochondrial complexes in *Plasmodium falciparum*”, Evers et al., have addressed queries from this reviewer satisfactorily.

Especially with regards to the oligomeric status of CV, the authors have now carried out extensive native mass calibration studies with commercial native markers as well as with bovine heart mitochondria, to demonstrate that the native complexes from mitochondria do not match in electrophoretic mobility

with the calibration markers. I agree that the detergent extracted parasite mitochondrial complexes compare well to the detergent treated BHM extracts, and this likely provides an excellent way to determine the native mass more accurately. Thus the mass of CV dimer appears to be ~2.2 Mda. However, it is also evident that the presence of detergent and type of detergent greatly affects the mobility of native proteins (based on the BHM data). In this context, I am wondering whether the NativeMARK proteins will migrate differently in the presence of detergent as well. Nevertheless, BHM based calibration probably more accurately represents the true mass of Plasmodium CV. Further, this is also in agreement with the formation of higher order oligomeric CV complex that has now been conclusively demonstrated in the related apicomplexan parasite *Toxoplasma gondii*.

The fact that the function for many of the novel proteins associated with CIII, CIV and CV are yet to be deciphered provided rich scope for further experimental work with these proteins & complexes; from structural and biochemical perspective. Moreover the presence of orthologs for many of these novel proteins in other apicomplexan and myzozoan species indicates conserved function and provides the opportunity to further study the evolutionary acquisition and retention of these proteins.

We thank the reviewer for the kind words and share his curiosity on how NativeMARK proteins will interact with detergents and whether this adaption would lead to better representation of proteins solubilized in the respective detergent. It may represent an interesting option for investigating compartments that are not as well represented by bovine heart mitochondria.

Minor Correction:

Table-I: for CV complex, the calculated monomer / dimer mass value is aligned to column-2 instead of column-3.

Changed accordingly

The manuscript is now significantly updated with new information, which clarifies many of the original queries raised. I now recommend accepting the manuscript for publication.